# CoCoMET v1.0: A Unified Open-Source Toolkit for Atmospheric Object Tracking and Analysis

Travis Hahn[1,*], Hershel Weiner[2,*], Calvin Brooks[3], Jie Xi Li[4], Siddhant Gupta[5], and Dié Wang[6]

[1]Department of Statistics, The Pennsylvania State University, University Park, PA 16802
[2]Department of Physics and Astronomy, University of Hawaii at Manoa, Honolulu, HI 96822
[3]Physics, Applied Physics, and Astronomy Department, Rensselaer Polytechnic Institute, Troy, NY 12180
[4]Applied Mathematics & Statistics, Stony Brook University, Stony Brook, NY 11794
[5]Environmental Sciences Division, Argonne National Laboratory, Lemont, IL 60439
[6]Environmental and Climate Sciences Department, Brookhaven National Laboratory, Upton, NY 11937
[*]These authors contributed equally to this work.

**Correspondence:** Dié Wang (diewang@bnl.gov)

**Abstract.** Advances in performance and analysis capabilities have accelerated the development of object tracking algorithms for atmospheric research. This has resulted in a growing number of studies using Lagrangian tracking techniques to analyze the evolution of atmospheric phenomena and the underlying processes. However, the increasing complexity and variety of tracking algorithms present a steep learning curve for new users and make it difficult for existing users to compare algorithm

performance.

We introduce CoCoMET (Community Cloud Model Evaluation Toolkit), an open-source toolkit that addresses these issues. CoCoMET simplifies the process of running multiple tracking algorithms simultaneously and analyzing objects in both model and observational datasets by specifying parameters in a single configuration file. It standardizes input data from different sources into a consistent format and unifies the tracking output across algorithms. CoCoMET enhances the functionality of

existing tracking methods by calculating additional properties such as cell growth and dissipation rates, perimeter, surface area, convexity, and irregularity. In addition, CoCoMET includes a novel method for identifying mergers and splits in 2D and 3D tracks and supports the integration of Eulerian/stationary datasets external to the tracking data for process studies. Its potential utility is demonstrated through examples of model intercomparison, model evaluation against observations, and comparisons between tracking algorithms. Designed for open-source environments, CoCoMET will continue to expand with

future releases, incorporating more input data types and tracking algorithms.

# 1 Introduction

For decades, object-tracking algorithms based on a Lagrangian framework have been used to identify and study meteorological phenomena. These algorithms enable users to link objects along their trajectories, allowing for detailed analysis of their evolution over time, which accelerates the process-level understanding of tracked systems (e.g., Leese et al., 1971; Dixon and Wiener, 1993; Johnson et al., 1998; Machado et al., 1998; Fiolleau and Roca, 2013). In recent years, the atmospheric sciences community has widely embraced these tracking algorithms, thanks to the growing availability of open-source tools (Table 1). The types of meteorological phenomena they are capable of tracking are getting broader as well, including individual convective clouds, mesoscale convective systems (MCSs), tropical and extratropical cyclones, atmospheric rivers, and equatorial waves (e.g., Prein et al., 2023).

Tracking algorithms often share similarities in certain aspects but differ significantly in others, as they might be originally designed to track different types of phenomena. These differences are particularly evident in the trackers' input data requirements, thresholds, and internal modules, which handle essential tasks such as object segmentation, linking objects across time steps, identifying splits and mergers, and enabling three-dimensional tracking (e.g., Prein et al., 2024; Feng et al., 2024). The performance of these algorithms can be optimized by applying them to datasets tailored to their specific modules, functions, and tuning parameters. For instance, the Tracking and Object-Based Analysis of Clouds (*tobac*) algorithm (Sokolowsky et al., 2024) has shown robust performance in tracking individual or isolated convective clouds (e.g., Oue et al., 2022; Gupta et al., 2024). In contrast, the Python FLEXible object TRacKeR (PyFLEXTRKR; Feng et al., 2023) was developed for tracking MCSs (Feng et al., 2021; Hayden et al., 2021; Cui et al., 2024; Robledo et al., 2024). Meanwhile, more versatile algorithms, such as the Multi-Object Analysis of Atmospheric Phenomena (MOAAP) algorithm (Prein et al., 2021), are designed to track a broad range of features, including MCSs, atmospheric rivers, and synoptic troughs.

Variations between algorithms can pose significant challenges for users trying to select the most suitable tracker for a specific environmental region or scientific question. Identifying the optimal tracker often demands a deep understanding of each algorithm's nuances, strengths, and limitations. While users may rely on suggestions from previous studies, it is more pedagogical to compare the results produced by different trackers and make an informed decision (e.g., Prein et al., 2024; Feng et al., 2024). However, this process is often prohibitively challenging and requires collaborative actions due to the significant computational resources needed and effort required to understand, install, and operate multiple trackers—particularly as new trackers continue to emerge. Unfortunately, familiarity with one tracker rarely translates to ease of use with another. Given that no single tracker can be perfect for all applications, an ensemble tracking approach would offer a more robust solution to mitigate discrepancies that may arise in downstream analyses since it helps account for the inequalities between different trackers.

Another significant challenge is ensuring consistency across data pre-processing, tuning parameters, tracking thresholds, and the calculation of tracked properties. Any mismatches in these aspects may introduce non-physical uncertainties, making direct comparisons of tracking results problematic. To address this, there is a clear need for an open-source toolkit that can

simultaneously run multiple trackers, unify thresholds, and standardize the calculation of properties associated with tracked features.

The use of object tracking for model evaluation is gaining popularity, both for model intercomparisons and for comparisons between models and observational data (e.g., Prein et al., 2024; Feng et al., 2024; Gilmour et al., 2025; Hahn et al., 2025a). Feng et al. (2024) evaluated various the DYnamics of the Atmospheric general circulation Modeled On Non-hydrostatic Domains (DYAMOND) model simulations of tropical MCS against satellite precipitation and brightness temperature products by using multiple different trackers. They reported that while the frequency of observed MCSs can have a spread of a factor of 2-3

across trackers, robust model evaluation can be achieved despite differences in the formulation of different trackers. In another tracker intercomparison, Prein et al. (2024) examined the sensitivity of MCS statistics from climate model simulations to the formulations of six different trackers. This work showed the use of different trackers can influence the conclusions drawn while evaluating model simulations against observations and that the frequency, size, and duration of tracked MCSs are highly susceptible to the tracker being used despite the use of consistent criteria to define an MCS.

Applying trackers to simulations from different models poses several challenges, including inconsistencies in model output variables and the calculation procedures for observable quantities. For example, the Weather Research and Forecasting (WRF) model (Skamarock et al., 2019) and the Regional Atmospheric Modeling System (RAMS) model (Cotton et al., 2003; van den Heever et al., 2023) differ in key aspects such as variable naming conventions, output hydrometeor classifications, and the methods used to compute essential properties like radar reflectivity or precipitation rate. Similarly, discrepancies between

models and observations often arise due to differences in available quantities, temporal and spatial resolution, or spatial coverage. To address these issues, a preprocessing step is crucial to standardize and reformat input data structures before running trackers on different inputs, ensuring compatibility and consistency.

In this study, we develop an open-source Python package, CoCoMET (Community Cloud Model Evaluation Toolkit), to streamline the pre-processing of model and observational data as inputs, enable simultaneous execution of multiple trackers,

and standardize the analysis of tracking outputs. One of the key highlights of this package is its simplicity, allowing users to perform all necessary tasks by editing a single configuration text file. This package also includes a newly developed function for detecting 2D and 3D merging and splitting events (Hahn et al., 2025a), which can be seamlessly integrated with various tracking algorithms. Besides, the 3D merging and splitting functionality rarely exists in existing trackers. Additionally, CoCoMET offers an important feature: the ability to link the system life cycle characteristics (tracking results) to their surrounding environmental

conditions (e.g., sounding data), which is particularly useful and recommended for the studies of aerosol-environment-cloud interactions (e.g., Veals et al., 2022; Wang et al., 2024). Finally, the ability to handle both model and observational data offers significant value to the community by reducing the distinct pre- and post-processing efforts required to evaluate model output and observational datasets. Often these efforts are fundamentally different from each other and pose another entry barrier for researchers who may specialize in either modeling or collecting observations and are looking to incorporate the other in their

analysis.

## 2 Overall Structure

CoCoMET operates by internally managing all data transformations and necessary variable calculations to prepare and execute various trackers, with users having the option to tune the corresponding parameters. It also computes key characteristics of tracked cells in a parallelized environment to enhance computational efficiency. A test with total elapsed run time of 487.0s in an unparalleled environment takes 53.0s in an paralleled environment (see CoCoMET/examples/Paper_plotting_nbs/parallel_ processing_time_analysis.ipynb). The framework consists of four main components: input pre-processing, tracker implementation, output unification, and analysis functions (Figure 1), while also providing the flexibility to incorporate additional modules and extend functionality within each component.

The atmospheric phenomena tracked by different algorithms are often addressed using different terminologies. CoCoMET encourages consistent definitions for an "object", "feature", and "cell" (Table 2). Any general atmospheric entity that is the subject of the tracking analysis is referred to as an "object". A "feature" refers to an object identified within a given 2D or 3D field at a single time step. A "cell" refers to a collection of objects identified across multiple time steps and linked to each other along a common trajectory by a tracker. This is consistent with the terminology used within *tobac* (Sokolowsky et al., 2024). Essentially, a cell represents the entire life cycle of an object identified within the input data more than once. The "life cycle" is primarily a property of cells since cells are a collection of objects identified across multiple time steps. However, a feature can also be associated with the "life cycle" of its parent cell if it is part of a collection of objects that make up that cell. While CoCoMET v1.0 was developed and validated for convective clouds as the tracked objects (Section 3), the architecture of the package remains adaptable to track any object represented as gradients within a 2D or 3D field.

CoCoMET is released under the BSD 3-Clause License on Github and can be installed using the Python package manager, pip (https://pypi.org/project/CoCoMET/). The package is platform-agnostic, although installation of CoCoMET can vary depending on user system specifics, such as available compilers. In the latest release (v1.0), Python versions 3.10 through 3.12 are supported, but this may be updated based on developments in CoCoMET's dependencies.

### 2.1 Input Pre-processing

CoCoMET can accommodate both observational datasets and model simulations as inputs for tracking features. Observational datasets supported by the current version of CoCoMET include brightness temperature from the Geostationary Operational Environmental Satellites (GOES), radar reflectivity from the Next Generation Weather Radar (NEXRAD) system, and any other gridded radar datasets. The required data format for the gridded radar datasets is explained in Appendix A. Additionally, the package provides a function to handle input data from multiple adjacent NEXRAD radars to track organized convective clouds that cover a large domain. The NEXRAD radar data gridding is performed using the Py-ART open-source package (Collis and Helmus, 2013). CoCoMET is designed with flexibility to integrate other gridded observational data streams within future releases. Potential ideas for future development of the package are identified in Section 4.

For model simulations, users can run built-in functions to precompute commonly tracked variables such as brightness temperature, precipitation rate, and radar reflectivity, if these variables are not directly available within the standard model output.

Currently, CoCoMET supports three model outputs: WRF, RAMS, and the non-hydrostatic mesoscale atmospheric model (MesoNH; Lac et al., 2018). The package is designed to facilitate the integration of additional models in future releases (Section 4) It is important to note that CoCoMET v1.0 does not execute numerical model simulations but rather uses the output files from model simulations provided by the user and formats them to create the input dataset for various trackers.

Given the variations in variable names, formats, dimensions, and sometimes even definitions across different models, each dataset undergoes a dedicated pre-processing step for standardization before being passed to the trackers. Users simply need to provide the model simulation data to CoCoMET and specify the field they want to track in a configuration file for the trackers. For commonly tracked variables such as vertical velocity, brightness temperature, precipitation rate, and radar reflectivity, standardized input names – "wa", "tb", "pr", and "dbz", respectively - are assigned regardless of the type of numerical model. Users can also track any variable in the simulations by specifying its original name in the configuration file. No additional steps are required to prepare input files. We detail the standardization process for each model output in the following sections.

### 2.1.1 RAMS

RAMS is a highly versatile numerical model developed at Colorado State University for simulating and forecasting meteorological phenomena. It consists of three main components: an atmospheric model for executing the simulations, a data analysis package for processing initial meteorological data, and a post-processing tool for visualizing and analyzing model output (Cotton et al., 2003).

The raw output data for each RAMS simulation case consists of two files: a netCDF file containing the numerical data for simulated variables and a text file storing metadata for the simulation (e.g., variable dimensions, model grid spacing). A more detailed description of RAMS can be found at https://rams.atmos.colostate.edu/detailed.html. A separate function in CoCoMET (`rams_configure.py`) is implemented to parse the metadata file and extract essential information required for tracking, such as the simulation start time, variable dimensions, grid spacings, and map projection.

Radar reflectivity is computed following the methods outlined in the RAMS source code (https://github.com/RAMSmodel/RAMS/tree/main) which assumes the hydrometeors (rain, pristine ice, snow, aggregates, graupel, hail) are spherical and their number concentrations are represented by gamma distributions. Brightness temperature is computed from the outgoing longwave radiation using the Stefan Boltzmann law (Yang and Slingo, 2001), and the same method also applies to WRF and Meso-NH simulations.

RAMS does not provide direct output variables for surface precipitation rate, a key variable commonly used for tracking convective cores in both isolated cells and MCSs. Therefore, CoCoMET includes a function to calculate surface precipitation rate based on additional model outputs. It can be calculated using different settings, determined by the user's input for `calculation_type` in the CoCoMET configuration file. If `calculation_type` is set to "surface time averaged precipitation rate" (default), the change in the sum of all surface accumulative hydrometeor mixing ratio rates (rain, pristine ice, snow, aggregates, graupel, hail, and drizzle) over consecutive model time steps is used for tracking. If set to "surface instantaneous precipitation rate", the sum of all surface hydrometeor rates simulated at each model time step is used for tracking. Finally, for

"volumetric instantaneous precipitation rate", the sum of all 3D hydrometeor rates derived at each model time step is used for tracking.

### 2.1.2 WRF

The WRF model is a widely-used mesoscale numerical weather prediction system developed for both atmospheric research and operational forecasting (Skamarock et al., 2019). It includes two dynamical cores, a data assimilation system, and a software framework designed for parallel computing and system expansion. The model is applicable to a broad spectrum of meteorological studies, covering spatial scales from a few meters to thousands of kilometers. The output of WRF is in netCDF format, containing both numerical variable data and metadata.

In addition to the direct outputs of WRF, which may be selected for tracking (e.g., updraft velocity and rain mixing ratio), we also implement functions to calculate radar reflectivity (when it is not directly output by WRF) using the methodology from Koch et al. (2005). The calculation is based on the `wrf.dbz` function in the wrf-python open-source package (Ladwig, 2017). The hydrometeors involved in the calculation include rain, snow, and graupel. In the next version of CoCoMET, we plan to replace the current equations used for calculating radar reflectivity with a radar simulator (e.g., The Cloud-resolving model Radar SIMulator; CR-SIM; Oue et al., 2020). This upgrade will extend the reflectivity calculation's applicability to simulations using various microphysical schemes (different hydrometeor types), enhancing CoCoMET's versatility.

As with RAMS, WRF does not provide direct output variables for surface precipitation rate. In CoCoMET, at a given time step, it is calculated using the methodology used to describe "surface time averaged precipitation rate" in RAMS, and is provided as the input "pr" to CoCoMET. It is estimated from the difference in accumulated precipitation between consecutive time steps. The accumulated precipitation is the sum of accumulated convective precipitation (RAINC) and accumulated grid-scale precipitation (RAINNC).

### 2.1.3 Meso-NH

Meso-NH is a non-hydrostatic mesoscale atmospheric model developed by the French research community (Lac et al., 2018). The model supports simulations across a wide range of scales, from large eddy to synoptic, with advanced physical parameterizations for clouds and precipitation (Taufour et al., 2024). Coupled with the SURFEX surface model (Schoetter et al., 2020), it represents surface-atmosphere interactions across different surface types. Meso-NH features grid nesting for multi-scale simulations, operates in 1D, 2D, or 3D, and it includes a chemistry module and a lightning module.

Meso-NH outputs data in netCDF format, similar to WRF, containing both meteorological variable values and associated metadata. Radar reflectivity is calculated in the same way as WRF inside CoCoMET with plans to upgrade using CR-SIM. Unlike RAMS and WRF, Meso-NH provides output variables for surface precipitation rate, which can be directly used for tracking precipitation cores. Since radar reflectivity is not a direct output from Meso-NH, users can calculate it using the same method described in the section above for WRF.

## 2.2 Implemented Trackers

The current version of CoCoMET supports three tracking algorithms — *tobac* (v1.5.3), MOAAP (v1.1.1), and TAMS (v0.1.5). These trackers were prioritized based on their openly accessible source codes, demand from collaborators and users, and their familiarity to the CoCoMET developers. Nevertheless, CoCoMET is designed with a flexible, modular structure (Figure 2) to facilitate the incorporation of additional open-source, Python-based trackers in future releases based on user feedback and research needs (Section 4). The independent integration of multiple trackers within CoCoMET ensures that the unique features of widely-used trackers are incorporated into the package. The open-source nature of CoCoMET enables users to suggest updates whenever an incorporated tracker releases a new version. Future versions of CoCoMET will integrate any updated tracker versions to ensure the package remains current.

### 2.2.1 *tobac*

*Tobac* is a Python-based, open-source framework designed to identify, track, and analyze atmospheric features in both 2D and 3D datasets. The package is designed in a modular framework with three main steps. In the first step, regions satisfying progressively restrictive thresholds (e.g., 30 dBZ, 40 dBZ, 50 dBZ; Gupta et al., 2024) in a given variable field are identified as 'features' (Table 2). Feature identification is followed by a 'segmentation' step wherein the area (for 2D datasets) or volume (for 3D datasets) associated with each feature is estimated at each time step. Finally, features identified across successive time steps are linked based on a search within a defined radius of their projected positions. *Tobac* is flexible as feature segmentation can be completed after either the identification or linking step, and can be bypassed if a user does not need spatial information for the tracked features. More specific details associated with each step are provided by Heikenfeld et al. (2019) and Sokolowsky et al. (2024).

*Tobac* v1.5.3 (Sokolowsky et al., 2024) introduced an increase in computational efficiency, tracking of 3D features, handling of feature splits and mergers, internal spectral filtering, and support for periodic boundary conditions, making it more robust for analyzing atmospheric data. *Tobac* has been widely used for tracking convective clouds, updrafts, precipitation systems, and other meteorological phenomena in both model simulations and observational datasets (e.g., Oue et al., 2022; Kukulies et al., 2023; Gupta et al., 2024). We ensure that unique elements of *tobac*, such as parallel computing and its ability to track any type of feature within a gridded field (even non-meteorological ones), are incorporated into CoCoMET. This tracker is integrated with all input data streams implemented in the current version of CoCoMET.

### 2.2.2 MOAAP

The MOAAP algorithm is designed to identify and track a wide range of atmospheric features within a unified framework. It processes both regional and global datasets to detect and analyze phenomena such as tropical and extratropical cyclones, mid-level cyclones, cut-off lows, anticyclones, atmospheric rivers, jet streams, fronts, MCSs, and tropical waves (Prein et al., 2023).

MOAAP begins by applying thresholding methods to classify grid points corresponding to specific atmospheric phenomena. It then generates label maps that delineate the spatial extent of each identified feature, with unique identification criteria tailored to each type of feature (e.g., vorticity for cyclones, moisture flux for atmospheric rivers). Once features are identified, MOAAP employs a nearest-neighbor approach and spatial overlap techniques to track the movement of features across consecutive time steps. The linking process primarily relies on a simple connectedness principle, i.e. adjacency in space and time between detected features. Once features are initially detected, that information is fed to a two-pass binary connected-component labeling algorithm (Dillencourt et al., 1992) for linking. MOAAP accounts for features that merge, split, or dissipate during the tracking process, maintaining accuracy in complex scenarios. After tracking, MOAAP generates comprehensive outputs that include all detected features and their trajectories.

This tracker is currently integrated with only three model outputs (i.e., WRF, RAMS, Meso-HN) in CoCoMET, as it requires a set of variables (see https://github.com/AndreasPrein/MOAAP/wiki) to track atmospheric phenomena that are often unavailable in a single observational dataset.

### 2.2.3 TAMS

The Tracking Algorithm for Mesoscale Convective Systems (TAMS) is a Python-based tool which operates in a series of steps: identification, tracking, classification, and variable assignment. It is primarily used to analyze organized cloud systems in both observational and model data using Infrared brightness temperature (Tb) or cloud top temperature (Moon and Ocasio, 2024). TAMS can handle data from both structured and unstructured grids, which makes it adaptable for various types of input datasets.

By default, TAMS identifies cloud elements by applying the following threshold criteria: an edge Tb or cloud top temperature below 235 K (a free parameter in TAMS) within an area larger than 4,000 $km^2$ (a fixed parameter) and an embedded cold core with a Tb or cloud top temperature below 219 K (a free parameter) that covers an area larger than 10 $km^2$ (a fixed parameter) and is detected at least once during the lifetime of the MCS.

After identifying the cloud elements, TAMS uses backward linking to group cloud elements that show significant overlap with projected elements from the previous time steps, creating MCS tracks. TAMS classifies each tracked event into different categories based on the size, structure, and longevity of the tracked MCSs. The detailed explanation of each category can be found in Moon and Ocasio (2024). Note that this tracker is currently integrated with three types of model outputs and GOES observations in CoCoMET.

### 2.3 Output Unification

Some variables are commonly provided as output by different trackers. These variables include feature IDs (identified at a specific point in time), cell IDs (the full spatial and temporal extent of a tracked system or a series of linked features across multiple time steps), cell location, start time, and end time, etc. However, how other cell characteristics are defined, computed, formatted, and stored (e.g., csv, NetCDF) can vary significantly across trackers. Additionally, not all trackers have the capability

to identify merging and splitting events. These inconsistencies present a significant challenge when analyzing and integrating results from different tracking algorithms.

To overcome this, CoCoMET reformats universally available output variables (e.g., cell location, timing) from various trackers in a consistent manner. It then standardizes the calculation of key cell properties (e.g., cell height and area) through standalone functions. This approach ensures consistency across different trackers and facilitates robust cloud lifecycle analysis.

Moreover, CoCoMET goes beyond traditional metrics by incorporating additional properties like perimeter and irregularity, providing a more detailed and comprehensive characterization of tracked atmospheric systems. Calculating these additional parameters can expand the convection tracking literature by elucidating key properties such as updraft shape, width, and size, while also addressing research questions related to cloud evolution, entrainment, and mass flux (e.g., Heiblum et al., 2019; Chen et al., 2023a). CoCoMET incorporates a newly developed method for identifying merging and splitting events in both 2D and 3D tracking (see Section 2.4), addressing a key limitation in many existing trackers. These functions can be executed either after initial tracking has been completed or specified in the configuration file to be applied automatically during the tracking process.

### 2.3.1 2D Perimeter

The perimeter of a tracked 2D feature (in km) is calculated by identifying and summing the lengths of its edge segments. First, the segmentation mask is used to identify the feature's boundary by checking each grid point in the feature. A point is considered an edge if at least one of its four neighboring points lies outside the feature. The length of each edge segment is calculated using the projection coordinates, then all edge segments are summed to determine the total perimeter. This process is repeated for each feature at each time step. The feature perimeter function is used extensively in the newly developed methodology within CoCoMET to identify splits and mergers between objects tracked in two-dimensional space.

### 2.3.2 Feature Area

The 2D feature area is determined through segmentation, resulting in a masked array that identifies grid points occupied by the feature at each time step. The area, expressed in square kilometers, is calculated as the product of the number of grid points within the feature and the area of an individual grid box. For 3D tracking, the feature area at a given height level can be calculated by specifying the height (2 km by default) within the configuration file in the segmentation function. This variable is important in multiple aspects of atmospheric sciences. For example, the extent of cloud cover directly influences Earth's energy balance (e.g., Hartmann, 2016; Sokol and Hartmann, 2020). Additionally, the size and distribution of precipitation areas determine the volume and distribution of rainfall, affecting water resource availability (e.g., Yu et al., 2022; Lee et al., 2023).

### 2.3.3 Feature Surface Area

In 3D tracking, the surface area of each feature is calculated by checking every grid point in the feature segmentation. For each point, the algorithm looks at its six connected neighboring points in the cubic orientation. If any of these neighbors is not part of the feature, the current grid point is considered "exposed", and the area of the exposed side is calculated using the projection coordinates. This process is repeated for all grid points, and the surface areas are summed up. The final result is in square kilometers and is stored for each feature in each time step for a tracked cell. The feature surface area function is used extensively in the newly developed methodology within CoCoMET to identify splits and mergers between objects tracked in three-dimensional space.

### 2.3.4 Feature Volume

The feature volume can be calculated when tracking cells using 3D data. It is determined by summing the volumes of all the grid boxes that define a feature according to the segmentation results. The final output is provided in cubic kilometers and is stored for each feature in each time step for a tracked cell. This variable has been frequently used to analyze cloud and precipitation characteristics. For instance, Prein et al. (2017) found that a warmer condition may increase the precipitation volume produced by MCSs by up to 80%.

### 2.3.5 Feature Irregularity

The irregularity or roughness of the edges of a 2D feature is quantified using convexity, a measure of how much a shape deviates from being convex. Convexity is defined as the ratio of the perimeter of the object's convex hull to the perimeter of the object itself. The convex hull is the smallest convex shape that fully encloses the object. A value close to 1 indicates a smooth, compact shape (Figure 3b), while lower values suggest jagged or complex boundaries (Figure 3a). This is useful in identifying cloud structures/morphology in meteorological data or model simulations and the potential influence by entrainment and mixing processes (e.g., Mellado, 2017; Chen et al., 2023b).

For a 3D feature, convexity is similarly used to evaluate the irregularity of an object's surface. Instead of using perimeter, the ratio is calculated using surface area—comparing the actual surface area of the object to the surface area of its convex hull. A value close to 1 indicates a compact, smooth structure, while a lower value suggests the object has indentations or rough, uneven surfaces. In both cases, convexity provides a quantitative measure of shape complexity of tracked clouds and/or system. Although such parameters have gained little popularity, they have the potential to facilitate studies, such as classifying different cloud fields (e.g., Lim and Daya Sagar, 2008) and determining whether an updraft is more thermal-like or plume-like (e.g., Morrison et al., 2020; Peters et al., 2020).

We also provide another way of quantifying the irregularity of a feature in 3D, which is called sphericity. Sphericity is a measure that compares the surface area of a perfect sphere with the same volume to the actual surface area of the feature being analyzed.

Mathematically, sphericity ($\psi$) is given by the equation:

$$\Psi = \frac{\pi^{\frac{1}{3}}(6V)^{\frac{2}{3}}}{A} \tag{1}$$

, where $V$ is the volume of the feature, $A$ is the surface area of the feature.

In Equation (1), the numerator represents the surface area of a sphere with the same volume as the feature, and the denominator is the actual surface area of the feature. By comparing these two quantities, sphericity provides a way to quantify how close the feature is to a perfect sphere, with values approaching 1 for more spherical shapes and values much less than 1 for highly irregular shapes. The sphericity is calculated for each feature in each time step for a tracked cell.

### 2.3.6 Maximum Height of a Variable of Interest

The maximum height of a variable is defined as the 95th percentile of altitudes (in km above ground level) where the selected variable exceeds a specified threshold across the entire segmented feature area.

For 3D tracking, this calculation is straightforward on a 3D variable of interest, as height data is directly available within the segmented feature. For 2D tracking, if the variable of interest is 3D with a vertical dimension, the maximum height can still be determined. The process involves examining the vertical columns within the horizontally tracked feature area and identifying the highest altitudes where the variable exceeds the given threshold (an independent threshold from tracking and segmentation).

For example, if tracking is performed based on updraft at 6 km altitude, and a threshold of 5 $ms^{-1}$ is given for the calculation of the maximum height, the maximum height of the updraft corresponds to the highest altitude (95th percentile) on the same 2D area as the segmented feature where the updraft velocity surpasses 5 $ms^{-1}$. The strength of convective clouds or the depth of convective cloud cores is often represented using parameters such as the Echo Top Height (e.g., Wang et al., 2019, 2024) which can now be derived automatically using CoCoMET.

### 2.3.7 Cell Growth and Dissipation Rates

The cell growth and dissipation rates are determined by calculating the rate of change in a targeted cell property (e.g., area for 2D tracking, volume for 3D tracking). For each cell, this property is taken at each frame/time step where the cell exists. The difference in the cell property between consecutive frames is divided by the time interval between the frames to estimate the rate of change in cell property. The cell growth/dissipation rate for the last feature along an object's life cycle are left as NaN. Such estimates can help understand temporal changes in cloud updraft width and the role of entrainment to determine its influence on convective cloud depth (e.g., Varble et al., 2024).

### 2.3.8 Feature Propagation Velocity

The propagation speed of a tracked cell is determined by estimating its position change over consecutive time steps. For each feature, the velocity is calculated as the displacement of the cell between its current frame and its next frame divided by the time interval. The velocity is output as a unit vector including propagation speed along two $(y, x)$ or three dimensions $(z, y, x)$.

The speed ($ms^{-1}$) is output as the magnitude of the original velocity vector. The velocity vector and speed for the last feature along an object's life cycle are left as NaN.

These quantities are particularly useful for examining the relative influence of local and synoptic-level forcings on cloud fields by comparing the propagation speed and direction of individual clouds against the background wind speed and direction (Corfidi, 2003; Zhang et al., 2021).

## 2.4 Identification of Mergers and Splits

Mergers and splits in cloud systems are key to understanding weather development and evolution (e.g., Westcott, 1984; Lu et al., 2022). A merger (or merging event) occurs when two or more individual tracked cells come together and are subsequently identified as a single cell in the next time step. A split (or splitting event) occurs when a single tracked cell divides into two or more distinct cells in a later time step. Mergers can combine smaller cells into larger, more organized storm systems that may lead to increased rainfall or severe weather, while splits can fragment these systems, changing the distribution and intensity of precipitation. Grasping these processes is essential for enhancing weather forecasts and refining Earth system models by improving our understanding of atmospheric dynamics like stability, turbulence, and energy transfer. CoCoMET implements a novel technique for identifying merge and split events during the lifecycle of tracked systems. As not all tracking methods include this capability, integrating a standardized procedure within CoCoMET ensures consistency in detecting such events across different trackers.

### 2.4.1 Mergers and Splits in 2D

The identification of mergers and splits follows the methodology of Hahn et al. (2025a), with modifications to improve adaptability to users' needs (Figure 4).

First, we Identify candidate pairs. Features that share a common border (referred to as "touching" features) exceeding a user-defined percentage of their perimeter are flagged as potential merging/splitting features. The default value is set to 20%. Next, we define the search region for feature area calculation. Each feature is approximated as a circle with a radius $r_{feature}$, which is calculated by adding up all "edge" grid cells (those which are not completely surrounded by segmented grid points) to define a circumference and divide this by $2 \times \pi$ to get the radius in "grid points." The radius is then expanded to rsearch using a user-defined weighting to define a search region for edge pixels for each feature. By default, the search region is set to 110% of the radius of each feature, making $r_{search} = 1.1 \times r_{feature}$. A square is created with the same center as the circle and with side length $2 \times r_{search}$. Within this square, each grid point is assigned a score based on its proximity to the feature core (point of maximum value) and its intensity relative to a background threshold. The calculation of the score ($S$) follows the formula below:

$$S_{i,j} = \omega_1 \frac{R_{i,j} - A}{R_{\max} - A} - \omega_2 \frac{\|(i,j) - (i_0, j_0)\|_2}{\sqrt{2}r_{\text{search}}}, \tag{2}$$

where $R$ represents the variable field, $R_{max}$ is the variable value at the feature core, $i_0, j_0$ is the location of the feature core, $A$ is a new background threshold for the tracked variable, set by the user (20 by default for radar reflectivity input), and $\omega_1$ and $\omega_2$ present adjustable weights that allow users to fine-tune the relative influence of intensity versus distance in determining mergers or splits. The default values for $\omega_1$ and $\omega_2$ are both 1.

If the score for a given feature exceeds the threshold, which is by default 0, it is included as part of a new mask. After applying this process to both features in the pair, the resulting masks are overlaid, and the overlap area is calculated as a percentage of either feature's area mask. Pairs that exceed a certain percentage area overlap threshold (50% by default) move to the final step and are referred to as potential merging or splitting pairs.

For these potential merging or splitting pairs, we look forward in time over a user-specified number of time steps (2 by default). If one cell persists and the other does not, this is classified as a merge event. However, if both features exist (or both do not exist) in the subsequent time steps, no merging event is confirmed. In cases where neither feature exists anymore, meaning both features are no longer tracked, they may have moved closer to each other but dissipated without merging, or they could have merged into other cells along the way.

At the same time, for these potential merging or splitting pairs, we look backward in time over a user-specified number of time steps (2 by default). If one feature exists consistently across the time steps while the other does not, this is confirmed as a splitting event. However, if both features exist (or both do not exist) in these time steps, no splitting event is recorded.

Unlike the original approach in Hahn et al. (2025a), which focuses on radar reflectivity only, this version in CoCoMET allows tracking based on any user-selected variable. For variables with inverse behavior (e.g., Tb, where lower values reflect stronger convection), the calculation is adjusted by inverting the variable field before the calculations. Examples of merging and splitting events are shown separately in Figures 5 and 6, respectively.

The output of this function is stored in two Pandas DataFrames (two tables), which record the time of the event (merge or split) and the cell IDs involved. In the merger DataFrame, each row contains a list of the two frames during which the merger occurred, a tuple of the two cell IDs of the parent cells involved, and the cell ID of the merged cell. In the split DataFrame, each row contains a list of the two frames during which the split occurred, the cell ID of the split cell, and a tuple containing the cell IDs of the two child cells.

Overall, this approach provides a flexible and robust way to track structural changes in 2D convective cells while allowing users to customize parameters for different atmospheric variables.

### 2.4.2 Mergers and Splits in 3D

We apply a similar approach to identifying mergers and splits in 3D as described for 2D data, with adjustments to account for the third dimension.

To speed up the process, before identifying mergers and splits, the dataset is filtered to focus on cells in the time steps immediately before feature disappearance or immediately after feature appearance. These are the only two instances when a merge or split can occur, so only the cells identified at these time steps are analyzed. This filtering step also pre-classifies all cells as being involved in either a merge or a split, which will be used in the final step of the identification process.

Instead of perimeter, we use the surface area of 3D objects to identify touching features at a given time step. In other words, for each feature, we evaluate how much of its surface area is shared with an adjacent feature to determine the potential for a merge/split. Rather than modeling the feature area as a circle in 2D, we use a sphere with an adjustable radius, extended in three dimensions. Similarly to the 2D process, we generate a cube with the same center as the sphere and side length $2 \times r_{search}$. Within the search cube, each grid point is assigned a score calculated using a formula similar to Equation (2) below:

$$405 \quad S_{i,j,k} = \omega_1 \frac{R_{i,j,k} - A}{R_{\max} - A} - \omega_2 \frac{\|(i,j,k) - (i_0,j_0,k_0)\|_2}{\sqrt{3}r_{\text{search}}}, \tag{3}$$

where $k$ is the vertical dimension. If the score of a grid point exceeds a user-defined threshold, it is considered part of the new mask. The overlap volume between two masks is then calculated to assess whether a merge or split has occurred. If the overlap volume exceeds a user-defined percentage of either mask's original volume, the event is confirmed as either a merge or split, based on the classification from the initial step.

Note that *Tobac* has a separate function for identifying mergers and splits, while MOAAP and TAMS do not output information about them. *Tobac* applies user-defined thresholds for maximum allowed spatial and temporal separations between cells to be considered potential mergers/splits. This is then used in an implementation of Kruskal's algorithm to construct a minimum Euclidean distance spanning tree. This tree structure, formed over spatial and temporal domains, is what defines the merge or split events. Unfortunately, there are no directly comparable thresholds or parameters between the *tobac* and CoCoMET 415   algorithms that would allow for a fair comparison.

## 2.5   Linking Tracking Output with Other Datasets

### 2.5.1   Linkage to Eulerian Data Sets

A key enhancement in the CoCoMET package is the integration of Lagrangian and Eulerian datasets for studying clouds and other atmospheric systems. While cell tracking provides a Lagrangian perspective on the lifecycle of a tracked property, not 420  all atmospheric measurements are collected in a manner that is suitable for tracking. Many observations, such as those from the Atmospheric Radiation Measurement (ARM) user facility (Mather and Voyles, 2013), are collected at fixed locations with one dimension (time) or two dimensions (time, height). Examples include but not limited to vertical pointing cloud radars and lidars, aerosol measurements at the surface or along a tethered balloon system, and precipitation measurements (disdrometers). To fully leverage these diverse datasets, it is essential to link cloud lifecycle stages with time-height observations, enabling a 425  more comprehensive analysis of cloud structure and evolution (e.g., Gupta et al., 2024; Wang et al., 2024).

      To achieve this, we developed a function (`extract_arm_product`) that extracts Eulerian measurements at each time step of the tracked cells. This function aligns the tracking frames with their corresponding timestamps and matches them to the nearest available Eulerian data time. It also calculates the time difference (`time_delta`) between the tracked cell's timestamp and the Eulerian data time, where positive values indicate Eulerian data recorded after the tracked frame and negative values 430  indicate earlier measurements. Additionally, the function identifies the closest feature and cell to a specified ground-based

measurement site or location, providing their respective IDs along with the distance between the Eulerian measurement site and the nearest tracked feature in kilometers. Such capabilities can easily be extrapolated to include distance from observations collected using mobile platforms like research aircrafts or Uncrewed Aerial Vehicles (UAVs), which will be considered in future development.

### 2.5.2 Linkage to the Environmental Conditions

To study aerosol-cloud-environment interactions and related processes, it is essential to link cloud properties with their surrounding environmental conditions. One key dataset we incorporate is the ARM INTERPSONDE data (Fairless et al., 2021), from which we derive convective indices such as Convective Available Potential Energy (CAPE), Convective Inhibition (CIN), and low-level wind shear (0-5 km), following the methodology of Wang et al. (2020).

These indices can be calculated using parcel theory under different assumptions, considering both irreversible pseudo-adiabatic and reversible moist adiabatic ascents. In the pseudo-adiabatic process, we assume an undiluted parcel ascent while neglecting hydrometeor loading. Users can also choose to include ice-phase processes, which introduce additional buoyancy above the melting level due to latent heat release during freezing. The function also allows users to specify initial parcels, with choices among the most unstable parcel, surface parcel, and mixed layer parcel. The surface-based parcel is defined as the parcel at the lowest sounding data level; the most unstable parcel is defined as the parcel that has the greatest virtual temperature in the lowest levels above surface (700 mb as default and can be changed by users); the mixing-layer parcel is defined as the parcel with properties of the mean of the user-defined boundary layer (500 m as default).

In summary, the following properties are unique to CoCoMET: 2D Feature Perimeter, Feature Surface Area, Feature Irregularity, Maximum height of a Variable of Interest, Cell growth and dissipation rates, Feature Convexity and Sphericity, Linking to ARM Datasets, Calculation and linking to sounding data.

## 3  Come to Practice

### 3.1  Configuration File

One of the key features of CoCoMET is its ability to seamlessly run multiple trackers or process multiple input datasets simultaneously with a single line of code or editing a single configuration file (referred to as CONFIG). Users only need to specify the necessary parameters in the CONFIG. This file defines essential settings for CoCoMET, including selected trackers, input data directories, and parameters for subsequent cell analysis.

Additionally, users have the flexibility to customize which cell properties will be included in the output. If certain cell properties are not specified in the CONFIG, users have the flexibility to compute them after running CoCoMET by calling the corresponding functions. This allows for a more streamlined workflow, enabling users to focus on essential parameters during the initial run while retaining the option to derive additional properties as needed. This approach enhances adaptability, ensuring that users can tailor their analysis without rerunning the entire tracking process.

Each tracker's inherent parameters are fully configurable, allowing users to fine-tune their settings based on specific research needs. To enhance computational efficiency, CoCoMET supports parallel processing, enabling faster data processing when multiple cores are available.

The CONFIG can be stored in either a `.yaml` format or as a Python dictionary object. This enables easy reproducibility and dissemination of the tracking and analysis setup. Example `boilerplate.yml` files are provided in the CoCoMET repository and also in Weiner et al. (2025). A detailed breakdown of each parameter and its function within the CONFIG is summarized in Table 3 and Figure 2.

## 3.2  Output Formatting

Another key feature of CoCoMET is its standardized output structure, which ensures that outputs from all trackers can be uniformly converted for additional analysis and intercomparison. The output from any given tracker is a Python object—an instance of a class containing data and methods—with three components: feature identification (returned as a GeoDataFrame), linking (returned as a GeoDataFrame), and segmentation (returned as an Xarray Dataset). Each row in a GeoDataFrame corresponds to a single detected feature, maintaining a consistent format across different tracking methods. Additionally, a single Python object of class `Analysis_Object` is returned to the user allowing for the post-hoc calculation of additional analysis variables if desired. This uniform structure enhances the ease of subsequent analyses, allowing users to compare results across different tracking approaches efficiently.

The performance of CoCoMET is limited by its dependencies, the user's machine, and the user's tracking goals. Initial input and tracking speeds are largely dictated by individual dependency performance, but speedups–such as multithreading from dask–are used when offered. Large datasets may be exceptionally slow, or even fail to run, if the user's machine does not have sufficient memory or other processing power. However, it is often the case that the tracking setup itself is the issue. For instance, the tracking of cell updrafts–where hundreds or thousands of features may be identified–are going to slow down the analysis module of CoCoMET significantly due to the high computational complexity of the analysis algorithms. We recommend to run CoCoMET, initially, without the analysis module to ensure your configuration parameters are set correctly and there are not large amounts of undesired features.

To ensure the stability and reliability of CoCoMET, we have incorporated continuous integration into our GitHub workflow. This framework includes automated checks for code formatting using Black, documentation updates via pdoc, linting with pylint, and test execution through pytest.

As part of our testing strategy, we include a functional test that ensures the package runs correctly using our testing dataset (Hahn et al., 2025b), and a corresponding pre-run "ground truth" output. By comparing CoCoMET's output against this ground truth, we verify that core functionalities are preserved across code updates and changes. To facilitate debugging, the verbose flag is enabled during these tests, allowing clear identification of potential breaking points. In addition, we implement a set of unit tests for the analysis module to ensure accuracy and reproducibility of computed diagnostics.

### 3.3 Examples using CoCoMET

 #### 3.3.1 Model Intercomparison

CoCoMET facilitates model intercomparison studies by enabling the comparison of different numerical model outputs for the same event. An example of this is illustrated in Figure 7, where we compare tracked 3D updrafts from outputs of WRF and RAMS for a case that occurred on 19 June 2013 over the Houston region. The thresholds for defining an updraft are set at $3\ ms^{-1}$, $5\ ms^{-1}$, and $10\ ms^{-1}$, and we use *tobac* for the tracking process. The temporal resolution of the model outputs is 5 minutes, with simulations running for a duration of 4 hours from 1600 - 2000 local time. The details of this case and the corresponding model setups are presented in Marinescu et al. (2021).

We present histograms of eight selected updraft characteristics in Figure 7. The maximum volume, maximum updraft velocity, and maximum updraft height represent the maximum values of these variables captured during the cell's lifetime. The remaining variables are feature-based, meaning that the values for each individual feature within the cell are plotted. In general, both models exhibit similar statistics for most variables although small differences appear for some variables. While the scientific reasons behind these differences are beyond the scope of this study, our primary goal is to demonstrate the utility of CoCoMET in facilitating this type of comparison. The CONFIG is available in Weiner et al. (2025). After running this CONFIG, users will obtain outputs from *tobac* for three inputs simultaneously without additional steps.

#### 3.3.2 Model Evaluation

For model evaluation, conventional methods typically compare bulk properties of clouds or other atmospheric variables; however, there is an increasing trend over the past decade toward comparing properties at the individual cloud level, particularly through the analysis of cell life cycles. This approach not only provides a more detailed assessment of clouds but also proves invaluable when evaluating regional and Earth system models, especially in relation to the diurnal cycle of cloud initiation and evolution on a global scale.

We have utilized CoCoMET to evaluate multi-case ensemble simulations of sea breeze convection days over the Houston region, as demonstrated in Hahn et al. (2025a). This example highlights the effectiveness of CoCoMET in facilitating such comparisons.

In Figure 8, we provide another example of evaluating RAMS simulations used in Figure 7 with NEXRAD observations for the same case. *Tobac* is employed for tracking, with tracking performed on radar reflectivity at 2 km height. The thresholds for defining tracked features are set at 30, 40, 50 dBZ. The horizontal grid spacings of the NEXRAD data and the RAMS simulations are both 1 km. The temporal resolution of both datasets are similar, around 5 minutes. In Figure 8, both RAMS and NEXRAD show peak cell area and reflectivity during the mature stage of convection (around lifecycle bin 4), as expected (e.g., Gupta et al., 2024). The cell area growth rate transitions from positive in the earlier stage to negative in the later stage, which is consistent between observations and simulations for this particular case. These results give us confidence in the tracking method and the implementation in CoCoMET.

This example is achieved by specifying parameters in a single CONFIG (available in Weiner et al. (2025)), which highlights its streamlined workflow and allows for rapid analysis. This makes it a powerful tool for both single-case and multi-case evaluations with observational data, identifying key differences and similarities between the two, ultimately accelerating model evaluation and development. This time-efficient processing also ensures that comparisons can be made over extended periods, which is particularly valuable for long-term studies, such as those involving seasonal or annual model performance.

### 3.3.3 Tracker Intercomparison

Differences in tracked cell properties are expected when using different trackers due to the varying designs and underlying algorithms of these methods. Whether such differences significantly affect the conclusions drawn in studies related to the tracked systems is worth exploring further. One approach to quantify these differences is to track the same system using multiple trackers and analyze how variations in tracking results influence subsequent findings. This process can help improve the robustness of scientific conclusions.

CoCoMET is specifically designed to simplify this task by allowing users to configure and execute multiple trackers simultaneously through specifications in the configuration file. The toolkit returns results from all selected trackers in a standardized format, making it easier to compare and interpret differences.

We illustrate this process with an example in Figure 9, where we track brightness temperature from CONUS404 WRF simulations (Rasmussen et al., 2023) using two thresholds, 219 K and 235 K, with three trackers: TAMS, MOAAP, and *tobac*. This case occurred on 19 June 2013, and the trackers were applied over a 24-hour period. The CONUS404 simulations are performed using WRF version 3.9.1.1, with horizontal grid-spacing of 4 km and temporal resolution of one hour. Figure 9 shows the initiation location and timing (colors) of tracked cells for all trackers.

Overall, all trackers identify cells in the same general region, but *tobac* detects more cells compared to TAMS and MOAAP. This difference arises because TAMS and MOAAP are specifically designed to track large, organized systems, whereas *tobac* is more focused on identifying isolated convective cells. Compared to TAMS, MOAAP applies additional thresholds internally which limits the number of cases tracked. Despite the difference in the number of tracked cells, the initiation timings are well captured by all trackers, demonstrating consistency in some aspects across the methods. The CONFIG file for running CoCoMET is available in Weiner et al. (2025).

## 4 Future Development Plan

To enhance the functionality and applicability of CoCoMET, we propose the following key areas for future development:

### 4.1 Enhancing Support to New Trackers

We will integrate additional, existing tracking algorithms such as PyFLXTRKR, ATRACKCS (An algorithm for TRACKing Convective Systems), TempestExtremes (Ullrich and Zarzycki, 2017), and other actively maintained, open-source trackers listed in Table 1, to facilitate intercomparison among additional trackers and the use of model-ensemble approach to rep-

resent uncertainties in the subsequent analyses. We will integrate additional, existing tracking algorithms such as PyFLX-TRKR, ATRACKCS (An algorithm for TRACKing Convective Systems), TempestExtremes, and other actively maintained, open-source trackers listed in Table 1, to facilitate intercomparison among additional trackers and the use of model-ensemble approach to represent uncertainties in the subsequent analyses. These trackers, along with the trackers already supported by CoCoMET, were included in a recent MCS Tracking Method Intercomparison (Feng et al., 2024). Future development will address enhancements for existing trackers within the package, for example, we plan to incorporate multiple data sources (e.g., GOES plus Stage IV) to enable MOAAP analysis using observations. CoCoMET developers will update the package every 6 months to account for new releases of the existing and newly incorporated trackers. Depending on the number of CoCoMET users, a discussion forum will utilize the open source ecosystem to incorporate the community's suggestions into any major releases.

We are also exploring the development of a machine learning-based tracking algorithm to enhance cloud detection and tracking efficiency, accuracy, and adaptability. We have maintained flexibility in CoCoMET's development to allow integration of gridded data streams such as the Stage IV precipitation dataset (Lin and Mitchell, 2005) and the European Centre for Medium-Range Weather Forecasts Reanalysis version 5 (ERA5; Hersbach et al., 2020) in future releases. In the future, CoCoMET will also the integration of additional models such as the Simple Cloud-Resolving Energy Exascale Earth System (E3SM) Atmosphere Model (SCREAM; Donahue et al., 2024) and the ICOsahedral Non-hydrostatic model (ICON; partnership, 2024).

Additional trackers can be integrated into CoCoMET via a plug-in structure. To do this, users or contributors should follow these steps:

1. Create a New Directory: Set up a dedicated directory containing the tracking algorithm's core functions if the tracker cannot be called directly from the original publicly available link.

2. Format Input Data: For each supported input type (e.g., WRF, RAMS, NEXRAD, GOES), create a script named `<data_source>_<tracker_name>.py` that reformats the input into the required format for the new tracker. This ensures compatibility with the specific input requirements of the tracking algorithm.

3. Translate Tracker Output: Add a function to `tracker_output_translation_layer.py` to convert the output of the new tracker into CoCoMET's standardized format.

4. Update Tracker Wrapper: In `run_tracker_wrapper.py`, update the `_run_tracker_det_and_seg` function to wrap the new tracker's execution and ensure it produces standardized outputs. Also, implement a `_<tracker_name>_analysis` function to apply CoCoMET's analysis routines to the tracker output.

5. Register the New Tracker: Finally, update the `run_tracker` function in `run_tracker_wrapper.py` to invoke all relevant functions created for the new tracker.

Another key focus is the development of tracking methods that incorporate multiple atmospheric variables to provide a more comprehensive view of cloud and precipitation systems. Furthermore, we aim to implement multi-feature tracking (e.g.,

updrafts and precipitation) to better understand interactions between different atmospheric processes. On top of these, we plan
to develop tracking capabilities for unstructured grid inputs, such as those used in the Model for Prediction Across Scales
(MPAS) model (Heinzeller et al., 2016). The capability for linking tracked features to environmental conditions (Section 2.5.2)
can be extended in future releases to include environmental parameters derived from other datasets, such as ERA5 reanalysis,
which will enrich the analysis and fill the spatial coverage gap between the tracking results and single point measurements.

## 4.2 Enhancing Support for Pre-tracking Input Datasets

We will extend compatibility with additional ARM scanning radar datasets and radars from other agencies to improve cloud
and precipitation tracking. Additionally, we will integrate other global and regional observational precipitation datasets, such
as Stage IV precipitation data, to enable broader applications and better suit for global model evaluation. To accelerate the
tracking process and improve user accessibility, we will develop streamlined methods for downloading and preprocessing
various input datasets.

We also plan to incorporate additional model outputs, including those from ICON and SCREAM to further enhance Co-
CoMET's capability for facilitating model intercomparison studies (e.g., Marinescu et al., 2021). Support for remapping irreg-
ular grids will be included in future releases of CoCoMET to support models that output non-cartesian grids.

To add an additional dataset into CoCoMET, contributors should follow these steps:

1. Update User Interface Layer: Create a `run_<data_name>` function in `user_interface_layer.py` which han-
   dles the calling of all possible trackers for the new dataset. Then follow the pre-existing procedure for other datasets in
   the functions `CoCoMET_start`, `CoCoMET_start_multi`, and `CoCoMET_load`.

2. Calculate Data Variables: Create a `<data_name>_calculate_products.py` file to facilitate the calculation of
   DBZ, WA, TB, and PR if the variables do not already exist in the data.

3. Generate Iris Cube: Create `<data_name>cube.py` to facilitate the generation of an iris Cube for one data variable.

4. Create Load File: Create a `<data_name>_load.py` file to load the data into an xarray Dataset and reference the iris
   Cube generator and data variable calculation files.

5. Run Individual Trackers: Create `<data_name>_<tracker>.py` files for each possible tracker which can be used
   on the data to facilitate tracker parameterizations.

6. Reference New Files in Wrapper: Finally, import newly created files and all necessary functions in `run_tracker_wrapper.py`
   and ensure proper naming conventions of each file.

## 4.3 Enhancing Support for Post-tracking Analysis Datasets

Understanding the large-scale regime of tracked events/days is crucial for studying cloud-environment interactions and the
related studies. To achieve this, we will incorporate synoptic weather regime classification products from ARM (Wang et

al., 2022; 2024), based on self-organizing maps, to better link tracked cells to synoptic conditions. Additionally, we will link tracking results with low-orbit satellite data (such as the Cloud, Aerosol and Radiation Explorer EarthCare; Wehr et al., 2023) to assign lifecycle stages to clouds sampled at the time of the satellite overpasses at locations of interest. We will also integrate the European Centre for Medium-Range Weather Forecasts Reanalysis version 5 (ERA5; Hersbach et al., 2020), AIRS (Atmospheric InfraRed Sounder; Tian et al., 2019), or other datasets to extract environmental conditions for tracked clouds, further supporting studies on cloud-environment interactions.

## 4.4  Visualization Enhancements

We will improve visualization tools for tracking outputs, including interactive maps and 3D representations. Additionally, we will develop user-friendly interfaces for exploring and analyzing tracking results with linked environmental data.

## 5  Conclusions

CoCoMET is a Python-based open-source package designed to facilitate the evaluation of cloud properties in both observational data and model simulations. With the growing interest in understanding cloud lifecycle characteristics and improving model representation of convection, CoCoMET addresses the challenge of efficiently conducting standardized, object-based comparisons across different datasets and trackers. The toolkit integrates multiple cell-tracking methods and supports data from various models (e.g., WRF, RAMS, MesoNH) and observations (e.g., radar, satellite).

CoCoMET provides a unified framework for defining and calculating fundamental cell characteristics, such as cell strength, size, height, and lifespan, ensuring consistency across datasets and trackers. The package's modular design allows users to easily configure and analyze single-platform or multi-platform datasets using a single configuration file. It provides developers with the flexibility to add new functions and incorporate additional input data streams in the future. The design of CoCoMET emphasizes computational efficiency, making it possible to perform long-term analyses or ensemble-based studies on large datasets, which are often limited by computational expense.

Finally, this manuscript highlights several case studies, including a model intercomparison using simulations of convective cells over Houston, an evaluation of WRF simulations against ground-based scanning radar observations, and a comparison between different trackers using CONUS404 simulations. These examples demonstrate CoCoMET's ability to streamline workflow, quantify differences, and identify key patterns of tracked cell properties. The package accelerates model evaluation and development by providing a robust, scalable, and time-efficient solution for object-based analysis.

*Code availability.*

The source code for the CoCoMET v1 package is available at https://github.com/ASCENT-BNL/CoCoMET and https://doi.org/10.5281/zenodo.15090741 (Hahn et al., 2025b).

The configuration files for running CoCoMET for results in Section 3 are available at https://doi.org/10.5281/zenodo.15048051 (Weiner et al., 2025).

*Data availability.*

WRF simulation data used in Figures 5 and 6 can be downloaded from https://app.globus.org/file-manager?origin_id=0c079436-56af-11ed-b805-855d8beae885&origin_path=%2F&two_pane=false (Prein et al., 2022; Ramos-Valle et al., 2023).

Model outputs used in Figures 7 and 8 can be accessed in the U.K. CEDA JASMIN supercomputer following processes listed in this document: http://acpcinitiative.org/Docs/Instructions_Jasmin_Workspace_171011.pdf (Marinescu et al., 2021).

CONUS404 data used in Figure 9 can be downloaded from https://www.usgs.gov/data/conus404-four-kilometer-long-term-regional-hyd (Rasmussen et al., 2023).

## Appendix A: CoCoMET Gridded Radar Data Requirements

To use gridded radar data as input in CoCoMET, the data must meet the following requirements (an example is available in the GitHub repository under `/examples/example_radar_standardized.py`):

### A1  Data Format

- The data must be an xarray DataArray named "`reflectivity`".

### A2  Required Dimensions (in the following order)

- **time**: A `numpy.datetime64` list of radar scan times.
- **z**: A list of altitudes in meters above the radar, with attributes:

- `standard_name`: "`altitude`"
   - `units`: "`m`"
- **y**: A list of y indices (e.g., `[0, 1, 2, ..., 500]`).
- **x**: A list of x indices (e.g., `[0, 1, 2, ..., 500]`).

### A3  Required Coordinates (in no particular order)

- **proj_y** (follows `y` dimension): A list of y distances in meters from the radar (e.g., `[-25000, ..., 25000]`).
- **proj_x** (follows `x` dimension): A list of x distances in meters from the radar (e.g., `[-25000, ..., 25000]`).
- **south_north** (follows `y` dimension): A list of y indices, identical to the `y` dimension.

- **west_east** (follows x dimension): A list of x indices, identical to the x dimension.

- **model_level_number** (follows z dimension): A list of z indices (e.g., `[0, 1, 2, ..., 40]`).

- **altitude** (follows z dimension): A list of altitudes in meters above the radar, identical to the z dimension (attributes are not necessary).

- **lat** (follows `y, x` dimensions): An array of latitudes with attributes:

  - `standard_name`: "`latitude`"

  - `units`: "`degree_N`"

- **lon** (follows `y, x` dimensions): An array of longitudes with attributes:

  - `standard_name`: "`longitude`"

  - `units`: "`degree_E`"

## A4    Required DataArray Attributes

- `long_name`: "`Reflectivity`"

- `units`: "`dBZ`"

- `standard_name`: "`equivalent_reflectivity_factor`"

*Author contributions.* TH: Coding, conceptualization, validation, writing; HW: Coding, conceptualization, validation, writing; CB: Coding, writing; JXL: Coding, writing; SG: Writing; DW: Coding, conceptualization, funding acquisition, supervision, writing.

*Competing interests.* The contact author has declared that none of the authors have any conflicts of interest.

*Acknowledgements.* This project was supported by the U.S. Department of Energy (DOE) Early Career Research Program, Atmospheric System Research (ASR) program, and the Office of Workforce Development for Teachers and Scientists (WDTS) under the Science Undergraduate Laboratory Internships Program (SULI). This paper has been authored by employees of Brookhaven Science Associates, LLC, under Contract DE-SC0012704 with the U.S. Department of Energy (DOE). SG is supported by Argonne National Laboratory under U.S. DOE contract DE-AC02-06CH11357 and the ARM User Facility, funded by the Office of Biological and Environmental Research in the U.S 695    DOE Office of Science.

We would like to acknowledge Dr. Aryeh Drager from Brookhaven National Lab for his help with the implementation of RAMS outputs and the RAMS developer team at Colorado State University for providing RAMS simulations. We also acknowledge the deep convection

model intercomparison project (MIP) of the Aerosol, Cloud, Precipitation and Climate (ACPC) initiative for providing model simulations used in Figure 7 of the manuscript.

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

**Table 1.** List of algorithms commonly used to track convective clouds and precipitation

| Tracking Algorithm | Codebase | Reference |
|---|---|---|
| *tobac* | github.com/tobac-project/tobac | Heikenfeld et al. (2019); Sokolowsky et al. (2024) |
| TAMS | github.com/knubez/TAMS | Moon and Ocasio (2024) |
| cloudbandPy | github.com/romainpilon/cloudbandPy | Pilon and Domeisen (2024) |
| MOAAP | github.com/AndreasPrein/MOAAP | Prein et al. (2023) |
| PyFLEXTRKR | github.com/FlexTRKR/PyFLEXTRKR | Feng et al. (2023) |
| TempestExtremes | github.com/ClimateGlobalChange/tempestextremes | Ullrich and Zarzycki (2017); Ullrich et al. (2021) |
| ATRACKCS* | github.com/alramirezca/ATRACKCS | Alvaro et al. (2022) |
| TINT* | github.com/openradar/TINT | Raut et al. (2021) |
| simpleTrack* | github.com/thmstein/simple-track | Crook et al. (2019) |
| KFyAO* | doi.org/10.1594/PANGAEA.877914 | Huang et al. (2018) |
| CITA | N/A | Borque et al. (2014) |
| TOOCAN* | N/A | Fiolleau and Roca (2013) |
| ForTraCC* | N/A | Machado et al. (1998) |
| TITAN* | github.com/NCAR/lrose-titan | Dixon and Wiener (1993) |

*Denotes lack of a public codebase or lack of code updated within 12 months.*

**Table 2.** Glossary of key terms used in the tracking analysis

| Term | Definition |
|---|---|
| Object | Any atmospheric/meteorological entity that is the target for the tracking analysis. |
| Feature | A single object at any given time step. Features identified within 3D domains are 3D features, and features identified within 2D domains are 2D features. |
| Cell | A collection of objects that represent a single target identified across multiple time steps. |

**Table 3.** CoCoMET Configuration File Specifications

| Parameter | Type | Description | Notes |
|---|---|---|---|
| verbose | Boolean (bool) | Controls whether CoCoMET outputs text during execution. | True enables text output; False disables it. |
| parallel_processing | Boolean (bool) | Enables multi-core processing when available. | True enables processing; False disables it. |
| max_cores | Integer (int) | Specifies the maximum number of cores CoCoMET may use if `parallel_processing` is True. | Limited by system resources. |
| bounds | Array ([float, float, float, float]) | Sets spatial bounds for inputs. | Optional; Format: [min_longitude, max_longitude, min_latitude, max_latitude]. |
| path_to_data | String (str) | Path to input data files. | Supports glob-like patterns (e.g., "wrfout_d02*"). |
| path_to_header | String (str) | Path to RAMS metadata .txt files. | RAMS only; Supports glob-like patterns (e.g., "RAMS_meta*.txt"). |
| is_idealized | Boolean (bool) | Flag indicating if input data is from an idealized simulation. | Default is "False". |
| min_frame_index | Integer (int) | Minimum frame index to select a subset of input data. | Optional; 0-based, inclusive; Each frame corresponds to a single input file. |
| max_frame_index | Integer (int) | Maximum frame index to select a subset of input data. | Optional; 0-based, inclusive. |
| feature_tracking_var | String (str) | Variable used for feature tracking. | e.g., "dbz", "tb", "wa", "pr", or other variable names from the input data. |
| segmentation_var | String (str) | Variable used for segmentation. | Same options as `feature_tracking_var`. |
| calculation_type | String (str) | Specifies the type of precipitation calculation. | RAMS only; Options: "surface time averaged precipitation rate", "surface instantaneous precipitation rate", or "volumetric instantaneous precipitation rate". |
| gridding | Dictionary | Parameters for NEXRAD data gridding. | Optional; Uses Py-ART gridding functions. |
| Tracker | Dictionary | Specifies the tracking method (e.g., *tobac*). | Tracker-specific parameters need to be defined; see the CoCoMET user guide for details. |
| Analysis | Dictionary | Contains computed variables as keys and required parameters as values. | Post-processing variables and parameters must be specified; see the CoCoMET user guide for details. |

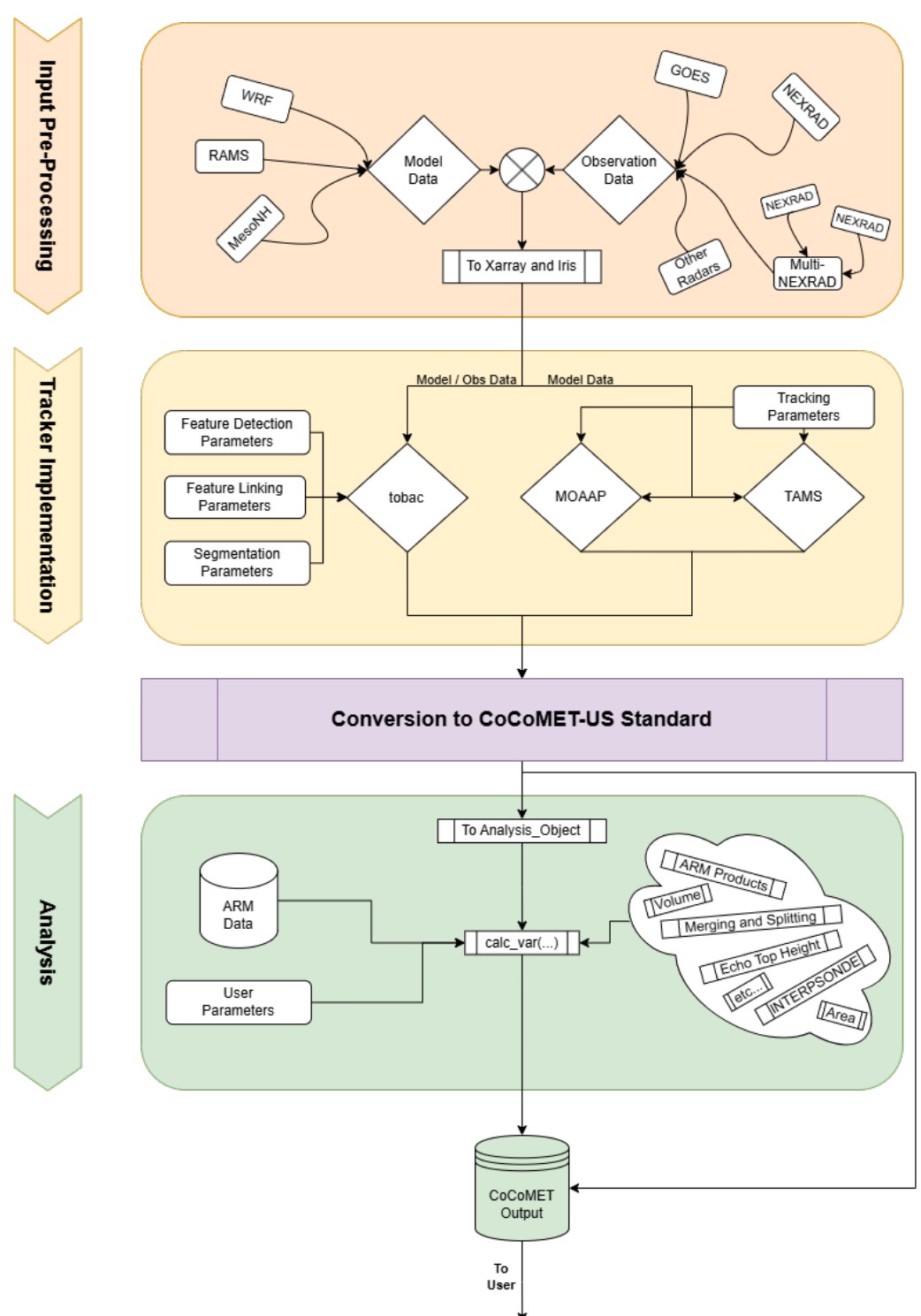

**Figure 1.** Flow chart highlighting the different components of the CoCoMET framework and the options available for user-based customization of parameters, input, and output fields.

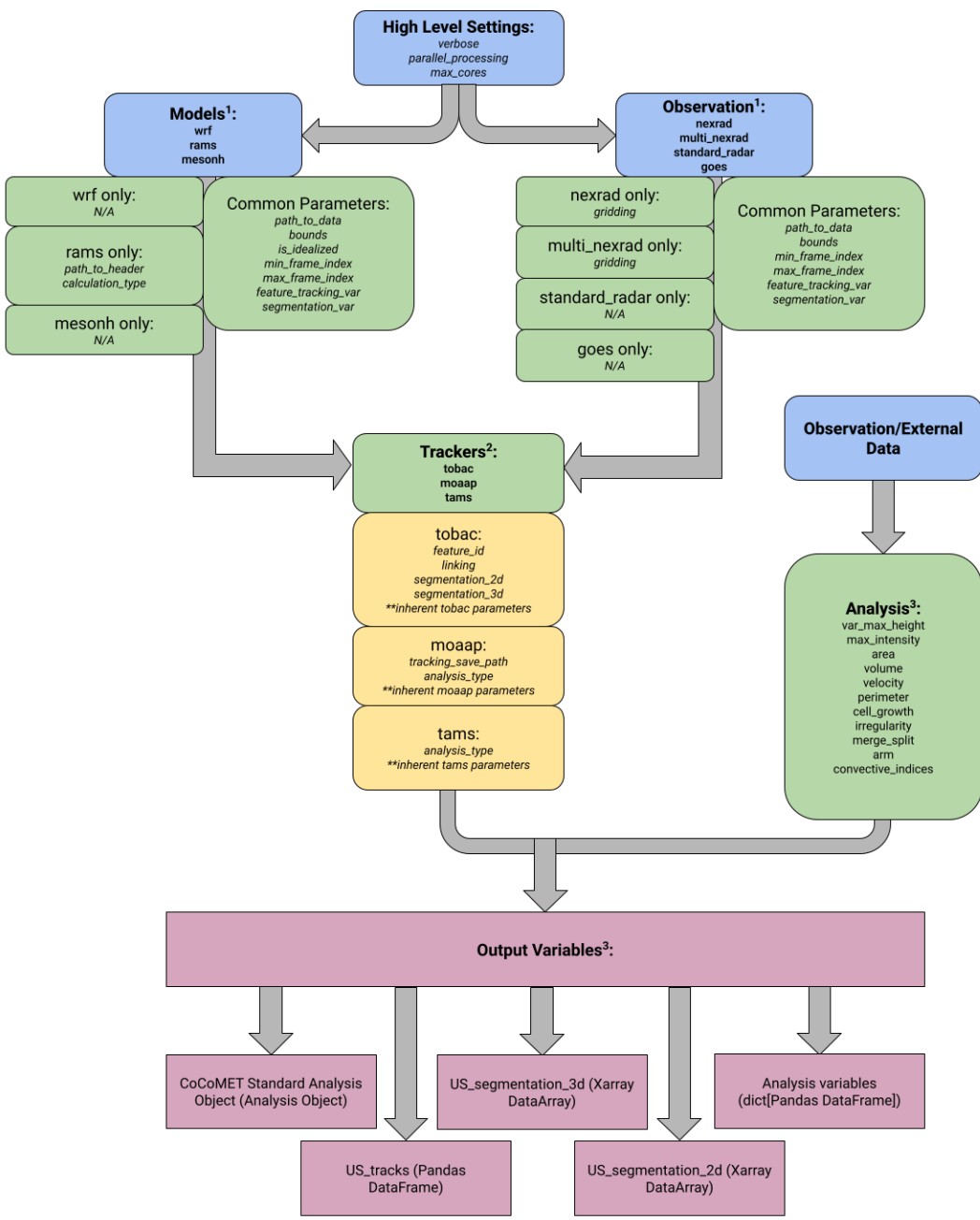

**Figure 2.** CoCoMET workflow highlighting the different input parameters for the configuration file, input data, and tracker options. Boxes marked as 1 are discussed in Section 2.1, as 2 are discussed in Section 2.2, and as 3 are discussed in Section 2.3.

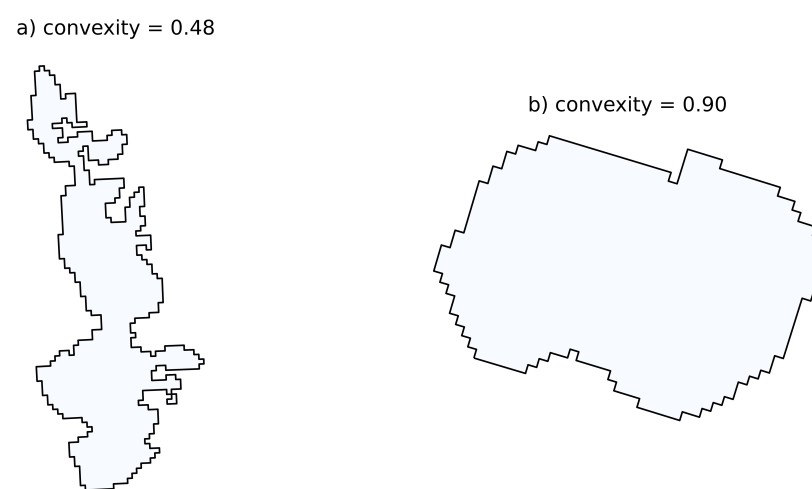

**Figure 3.** Illustration of convexities calculated for two tracked features.

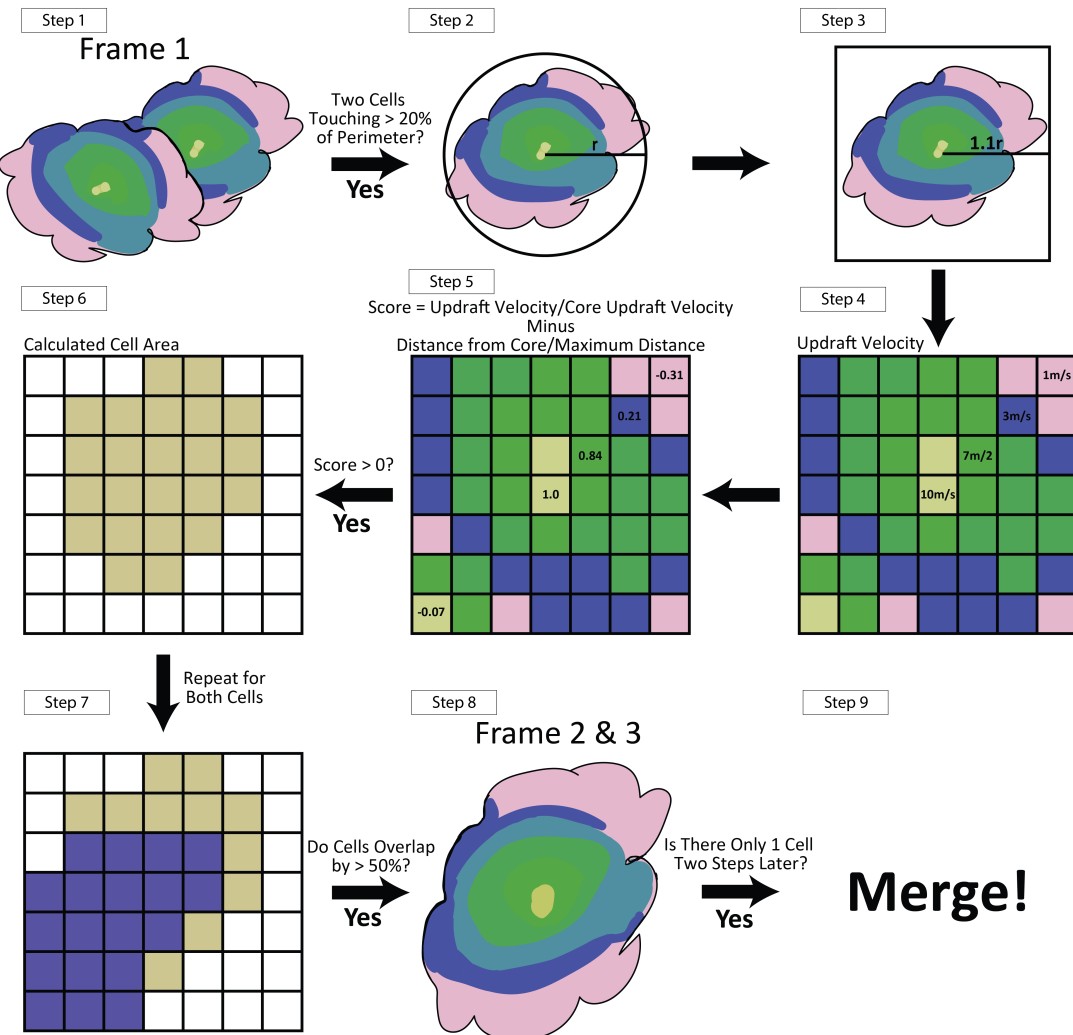

**Figure 4.** Illustration of steps for identifying mergers. An example input variable is updraft velocity at a certain height, using a threshold of $1\ ms^{-1}$. Colors from green to purple represent the updraft velocity or score, ranging from high to low. The dust color in step 6 represents the calculated cell grids for cell 1 and purple for cells 2 in step 7.

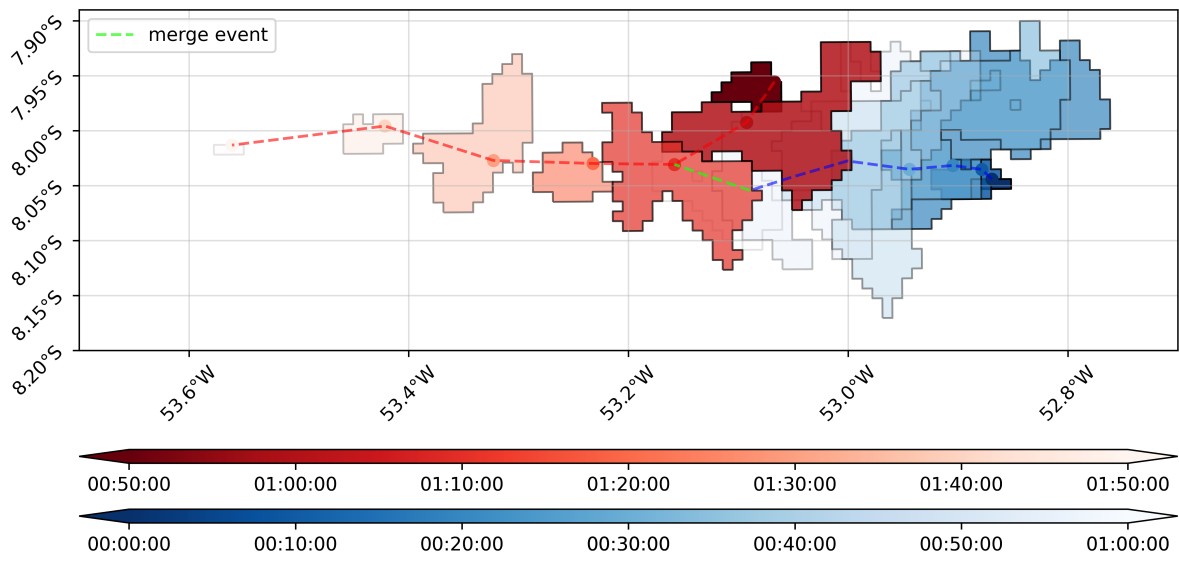

**Figure 5.** Example of a merging event detected by our new method using tracking outputs from *tobac*. The inputs for *tobac* are WRF simulations (Prein et al., 2022) of 2-km radar reflectivity at 1 km grid spacing for a deep convective case that occurred on April 1, 2014, in the Amazon. The tracking thresholds are 30, 40, and 50 dBZ. The cell in blue merges into the red one as both cells propagate toward the west.

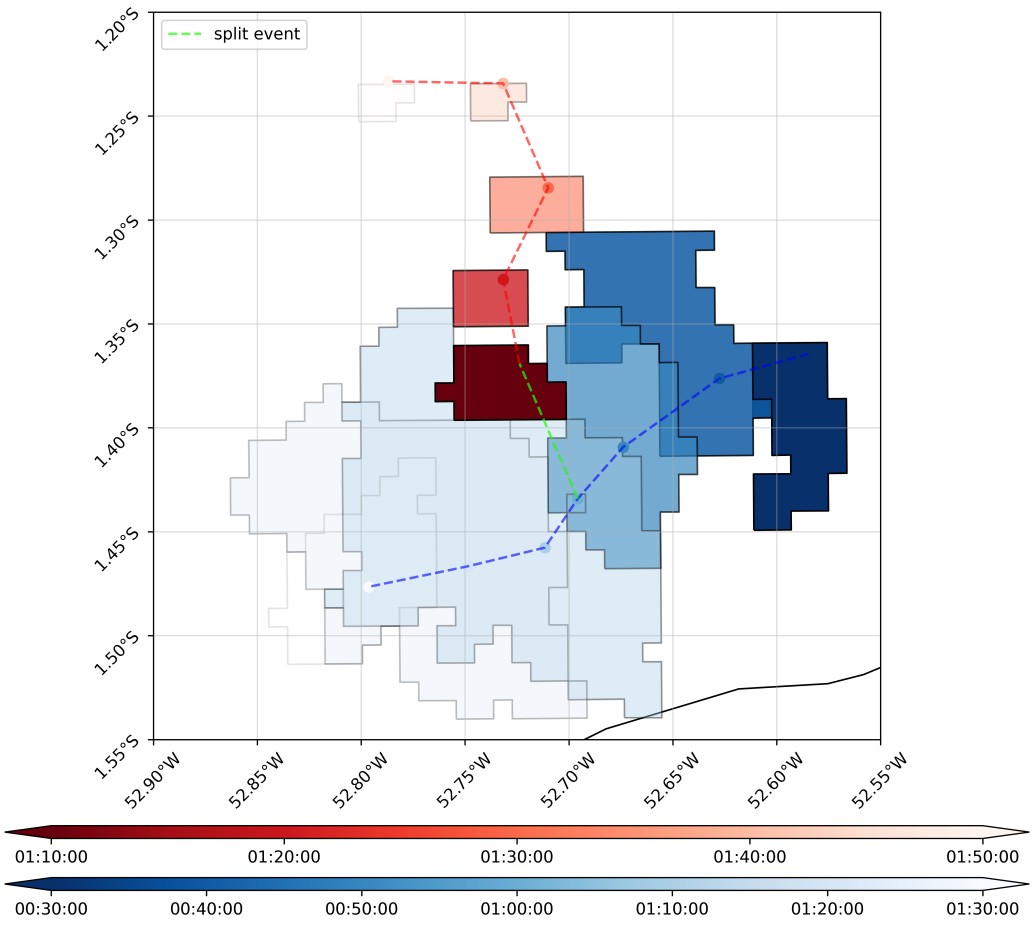

**Figure 6.** Example of a splitting event detected by our new method using tracking outputs from *tobac*. The inputs for *tobac* are WRF simulations (Prein et al., 2022) of 2-km radar reflectivity at 1 km grid spacing for a deep convective case that occurred on 1 April 2014, in the Amazon. The tracking thresholds are 30, 40, and 50 dBZ. The cell in blue splits into the blue and red cells.

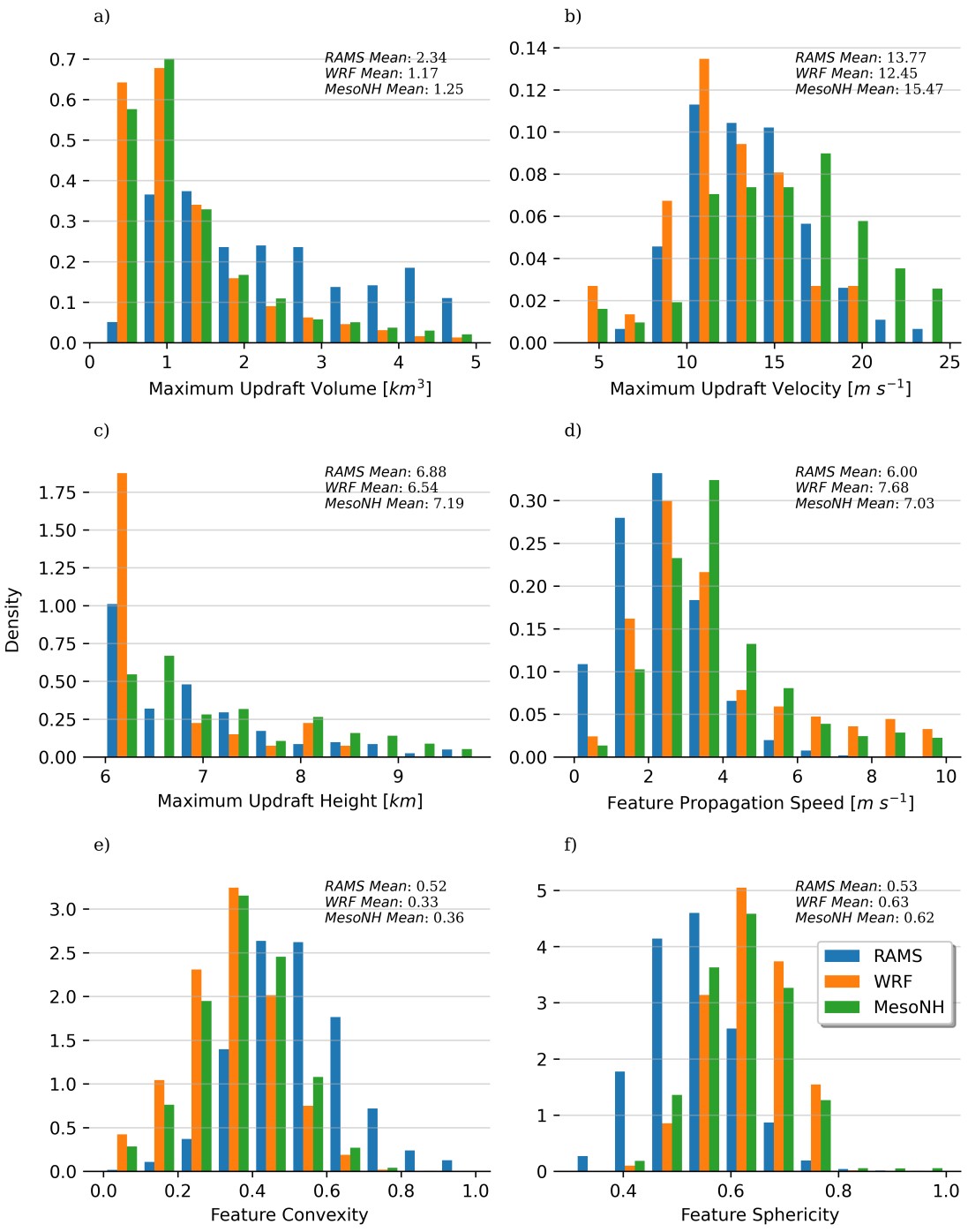

**Figure 7.** Histograms of tracked updraft properties using *tobac* based on simulations from WRF, Meso-HN, and RAMS for cases occurred on 19 June 2013 over the Houston region. b) and c) are plotted only for updraft cells that extend higher than 6 km to highlight those deeper cores.

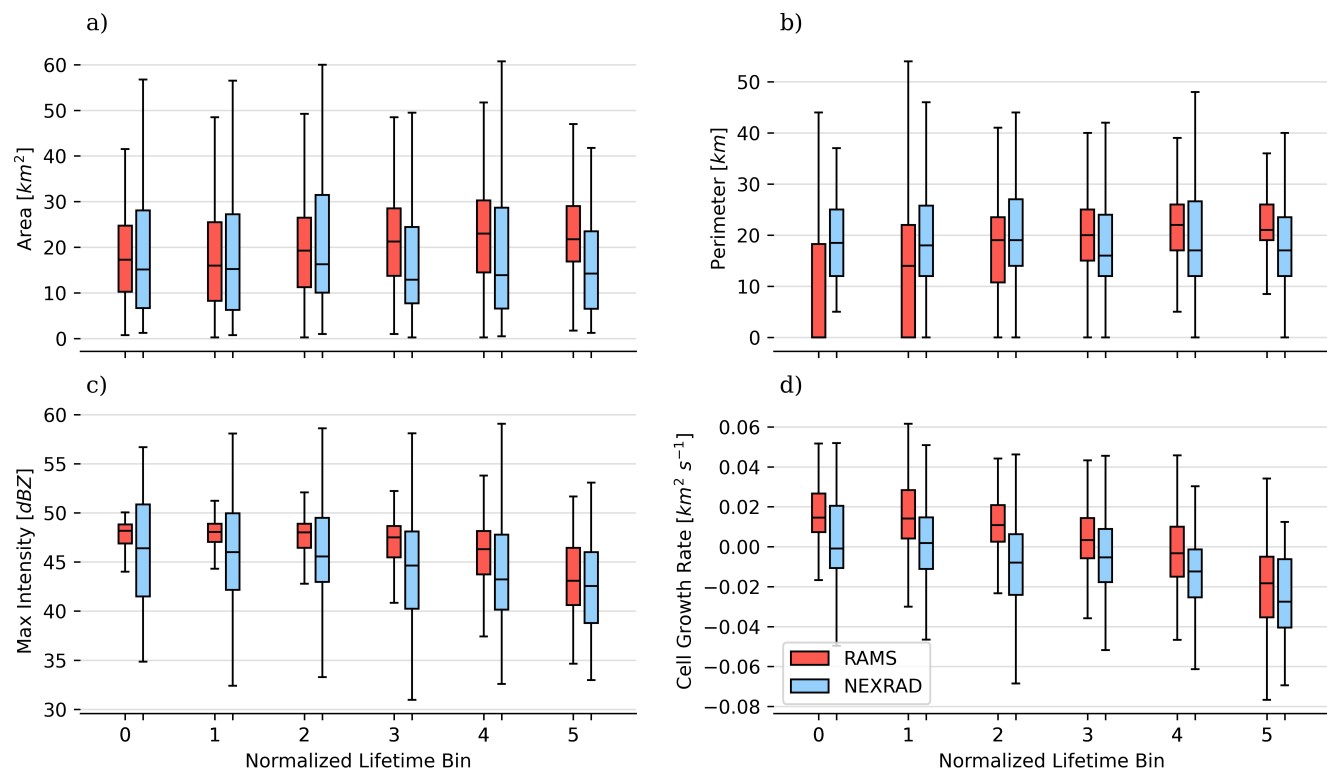

**Figure 8.** Box-whisker plot of tracked cell properties (both observed [NEXRAD] and simulated [RAMS]) as a function of the normalized cell lifetime bin. Only cells that last longer than 40 minutes and do not experience merging during their lifetime are included. 0 represents the first identification of the cell, and 5 indicates the termination of the tracked cell.

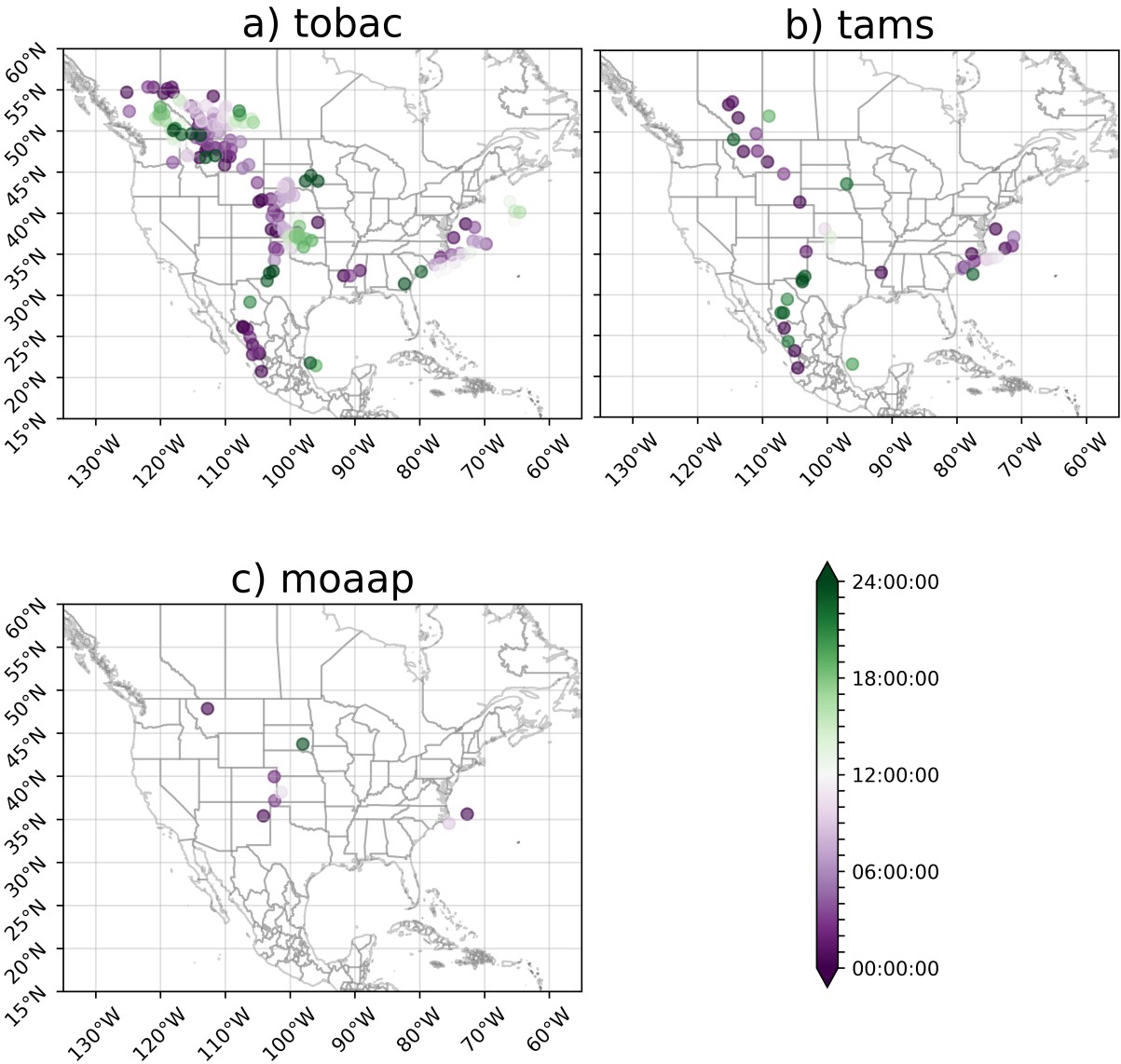

**Figure 9.** Initiation location and times (colors) for cells identified on 19 June 2013 in the CONUS404 WRF simulations using (a) *tobac*, (b) TAMS, and (c) MOAAP.