# Peer review of "CoCoMET v1.0: A Unified Open-Source Toolkit for Atmospheric Object Tracking and Analysis"

_EGUsphere, 2025_

## Author Response (AR2)

**Responses to Reviewers' Questions**

We thank both Reviewers for their insightful comments. We appreciate the opportunity to clarify and enhance our explanation and manuscript. Our responses are in blue.

**Reviewer #1**

The manuscript introduces CoCoMET v1.0, an open-source Python toolkit that ingests a variety of model and observational data sets, launches several established tracking algorithms (tobac, MOAAP, TAMS) from a single configuration file, and harmonises and enriches the resulting tracks with advanced diagnostics (perimeter, volume, convexity, 2-D/3-D merging-splitting convective cells detection, etc.).

Case studies demonstrate applications to model intercomparison, model–observation evaluation and tracker intercomparison. By unifying disparate trackers under a common interface and producing a standard, analysis-ready output, CoCoMET tackles a recognised bottleneck in object-based cloud research. The toolkit is potentially a valuable community resource and, once the revisions below are addressed, I expect it to be suitable for publication in GMD. Nevertheless, before publication the manuscript would benefit from a careful language edit before resubmission. Once these points are addressed I would be happy to recommend publication.

General comments:

Please state an SPDX-recognised licence (e.g. MIT, BSD-3-Clause, GPL-3.0), the minimum supported Python version and the recommended installation method (pip or conda).

We have included a reference to the BSD-3-Clause license, added the supported Python version, and specified the recommended installation method in Section 2 and also as follows:

*"CoCoMET is released under the BSD 3-Clause License on Github and can be installed using the Python package manager, pip (https://pypi.org/project/CoCoMET/). The package is platform-agnostic, although installation of CoCoMET can vary depending on user system specifics, such as available compilers. In the latest release (v 1.0), Python versions 3.10 through 3.12 are supported, but this may be updated based on developments in CoCoMET's dependencies."*

The manuscript needs a thorough language edit. Several forward-looking sentences currently placed in Sections 2 and 3 would read more naturally in Section 4 (Future development). Lines 87–95 and 404–406. These sentences discuss future extensions (e.g. extension to ERA5 data) and belong in Section 4.

We have gone through Sections 2 and 3 and moved any specific references of future development to Section 4. However, certain statements about package design have been left in Sections 2 and 3. This is to maintain proper flow of the discussion.

Although the text advertises "parallelised execution", no timing or scaling information is given. Even approximate wall-clock times and memory footprints for different cases would greatly assist prospective users.

The CoCoMET parallelized execution was tested on CoCoMET standard test data (Hahn, 2025) for each implemented tracking algorithm (tobac, MOAAP, and TAMS) using allubock's MicroBench Python package. In CoCoMET's parallelized environment, the total elapsed run time was 53.0s compared to an unparalleled environment which took 487.0s. The testing codes can be found in the GitHub under path: CoCoMET/examples/Paper_plotting_nbs/parallel_processing_time_analysis.ipynb. This info was added to Section 2.

Section 2.4 claims a novel merger–split method but offers only schematic examples; please provide quantitative validation (e.g. skill scores or comparison with tobac's own split/merge routine).

New developed function for detecting 2D and 3D merging and splitting: Could you provide an intercomparison between other's tracker merging and splitting and you method?

Tobac and CoCoMET have independent algorithms to track merging/splitting cells. Tobac applies user-defined thresholds for maximum allowed spatial and temporal separations between cells to be considered potential mergers/splits. This is then used in an implementation of Kruskal's algorithm to construct a minimum Euclidean distance spanning tree. This tree structure, formed over spatial and temporal domains, is what defines the merge or split events. In contrast, CoCoMET uses multiple criteria to identify cell mergers/splits, such as the percentage of spatial overlap between features and whether features share edges, which offers more flexibility. We added these texts to the revised manuscript.

Unfortunately, there are no directly comparable thresholds or parameters between the two algorithms that would allow for a fair comparison. Therefore, we performed our comparison using the default merge/split settings provided by CoCoMET and several settings for tobac (Figures below). It's important to note that these thresholds should be tailored to the specific application, as appropriate settings may differ between isolated and organized convection.

[Figure]

**Figure 1:** The locations (dots) and timing (colors) of mergers detected using both tobac (a) and CoCoMET (b) merging algorithms. The input data are infrared brightness temperatures calculated based on WRF simulations at 4 km resolution on June 19, 2013. Cells are tracked using tobac. For CoCoMET: touching_threshold = 0.15, score_weight_1 = 1.5. For tobac: merge_dist = 25 (100 km), frame_len = 7 (7 hours).

[Figure]

**Figure 2**: Same as Figure 1, but For CoCoMET: touching_threshold = 0.15, score_weight_1 = 1, overlap_threshold = 0.2. For tobac: merge_dist = 25 (100 km), frame_len = 3 (3 hours).

Output format – CoCoMET saves results in Python pickles, which are opaque outside the Python ecosystem and brittle across versions. Please justify the choice against an export option to self-describing format (e.g. NetCDF)

We have added an option to save CoCoMET output data as NetCDF files (pickles are kept as default). An example notebook is at 'CoCoMET/examples/saving_data_example.ipynb'. Additionally, users can save CoCoMET output data using the CoCoMET_save_output function, passing the output data and path as parameters to the function.

Terminology – The distinction between feature, cell and object is unclear. A short glossary or an early discussion (in Section 2) would help. Explicitly state that v1.0 is designed and validated for convective phenomena (and convective cells), even though the architecture could handle other objects.

Thank you for this suggestion, we have added the following table to the manuscript. Additionally, these terms are now discussed in plain language at the beginning of Section 2. We have added a comment on v1.0 being developed and validated with atmospheric convection in mind.

Table 2. Definitions for atmospheric phenomena in the context of tracking software.

| Term | Definition |
|------|------------|
| Object | Any atmospheric/meteorological entity that is the target for the tracking analysis |
| Feature | A single object at any given time step |
| Cell | A collection of objects that represent a single target identified across multiple time steps |

CoCoMET output format: The new toolkit CoCoMET highlights that it simplifies the run of trackers on different types of data, the analysis of trackers' outputs and their intercomparisons. However, most of the trackers output in the NetCDF format which is conveniently easy to use and process, and CoComet outputs have binary pickle format which can't be read without the CoComet specific routine. Please comment on whether this choice could limit long-term usability for users who rely on NetCDF.

Please see our answers to the above question.

Specific comments

Introduction (lines 55–64). The problem statement would be stronger if you cited concrete examples of prior tracker-intercomparison and model-evaluation studies.

Thank you for this suggestion. We have added the following text to the introduction:

*"The use of object tracking for model evaluation is gaining popularity, both for model intercomparisons and for comparisons between models and observational data (e.g., Prein et al., 2024; Feng et al., 2025; Gilmour et al., 2025; Hahn et al., 2025). Feng et al. (2025) evaluated various DYAMOND model simulations of tropical MCS against satellite precipitation and brightness temperature products by using multiple different trackers. They reported that while the frequency of observed MCS can have a spread of a factor of 2-3 across trackers, robust model evaluation can be achieved despite differences in the formulation of different trackers. In another tracker intercomparison, Prein et al. (2024) examined the sensitivity of MCS statistics from climate model simulations to the formulations of six different trackers which included tobac, MOAAP, and TAMS. This work showed the use of different trackers can influence the conclusions drawn while evaluating model simulations against observations and that the frequency, size, and duration of tracked MCS are highly susceptible to the tracker being used despite the use of consistent criteria to define an MCS."*

Introduction (74–79). Here the terms feature/cell are introduced but not defined; please add definitions and explain that CoCoMET v1.0 focuses on convective clouds and MCSs.

Please see our answers to the above question.

Also clarify the meaning of "life cycle" Which lifecycle are you referring to? The one of the "cell" of the "feature".

The "life cycle" is primarily a property of cells. This is because cells are a collection of objects (features) identified across multiple time steps and can therefore have a "lifetime". In contrast, a feature is just a single object at any point of time, so it has no primary property of "life cycle".

This definition has been added to the manuscript.

Introduction (closing paragraph). Highlight explicitly that CoCoMET can accommodate both observational and model data and summarize its community value.

We added: "*Finally, the ability to handle both model and observational data offers significant value to the community by reducing the distinct pre- and post-processing efforts required to evaluate model output and observational datasets. Often these efforts are fundamentally different from each other and pose another entry barrier for researchers who may specialize in either modeling or collecting observations and are looking to incorporate the other in their analysis.*"

L 90-95. Do you perform any grid remapping before applying the tracking? How does the package handle different grids as it would have for ICON or other models with non regular meshes?

The current version of CoCoMET does not support irregular grids. In the current version, a user needs to apply their own gridding to the irregular input dataset before using a tracker (except for NEXRAD data). The gridding functionality is planned for the next release. We have added the following sentence to Section 4.2: "*Support for remapping irregular grids will be included in future releases of CoCoMET to support models that output non-cartesian grids.*"

Section 2

Figure 2, together with the surrounding text leave it ambiguous whether CoCoMET executes atmospheric models. Please state explicitly whether model execution is in scope. Generally, in section 2, make it clear what belongs to models and what belongs to CoComet.

CoCoMET v1.0 does not directly execute any numerical model simulations. We added the following sentence to section 2.1 in the revised manuscript to address this ambiguity from the original manuscript.

"It is important to note that CoCoMET v1.0 does not execute numerical model simulations but rather uses the output files from model simulations provided by the user and formats them to create the input dataset for various trackers."

In Section 2.1.1 the final paragraph contains essential information but lacks context. A short lead-in sentence would help. In addition, please explain why precipitation rates are recomputed for RAMS and WRF (lines 137–140), whether these fields are required for identifying precipitation cores in MCS tracking, and, if so, consider adding a short dedicated sub-subsection on precipitation. At present this need becomes clear only later in the Meso-NH discussion (line 149).

RAMS and WRF do not provide direct output variables for surface precipitation rate, a key variable commonly used for tracking convective cores in both isolated cells (e.g., using tobac) and MCSs (e.g., using MOAAP).

We added these sentences: "RAMS does not provide direct output variables for surface precipitation rate, a key variable commonly used for tracking convective cores in both isolated cells and MCSs. Therefore, CoCoMET includes a function to calculate surface precipitation rate based on additional model outputs. "

We also added this sentence to the WRF section: "As with RAMS, WRF does not provide direct output variables for surface precipitation rate."

L139-140. Move variable names RAINC and RAINNC inside parentheses and move the explanatory text outside; re-phrase for clarity.

We rewrote the sentence: The accumulated precipitation is the sum of accumulated convective precipitation (RAINC) and accumulated grid-scale precipitation (RAINNC).

L 144- Add references for SURFEX and the cloud-microphysics schemes used in Meso-NH. Throughout, write "Meso-NH," not "MesoNH."

We added the references to the revised manuscript.

Section 2.2 (Implemented trackers)

Briefly explain how additional trackers can be integrated via plug-ins, so the community understands how to contribute.

Yes, additional trackers can be integrated into CoCoMET via a plug-in structure. To do this, contributors should follow these steps:

1. Create a New Directory: Set up a dedicated directory containing the tracking algorithm's core functions if the tracker cannot be called directly from the original publicly available link.

2. Format Input Data: For each supported input type (e.g., WRF, RAMS, NEXRAD, GOES), create a script named <data_source>_<tracker_name>.py (e.g., wrf_tobac.py) that reformats the input into the required format for the new tracker. This ensures compatibility with the specific input requirements of the tracking algorithm.

3. Translate Tracker Output: Add a function to tracker_output_translation_layer.py to convert the output of the new tracker into CoCoMET's standardized format.

4. Update Tracker Wrapper: In run_tracker_wrapper.py, update the _run_tracker_det_and_seg function to wrap the new tracker's execution and ensure it produces standardized outputs. Also, implement a _<tracker_name>_analysis function to apply CoCoMET's analysis routines to the tracker output.

5. Register the New Tracker: Finally, update the run_tracker function in run_tracker_wrapper.py to invoke all relevant functions created for the new tracker.

This plug-in approach is designed to be modular and extensible, allowing community contributors to integrate and test new tracking algorithms with minimal disruption to the core codebase. We have added these steps to the revised manuscript in Section 4.1.

L146, numerical variables or meteorological variables?

Yes, it should be "*meteorological variable values and associated metadata.*"

L146-147. Specify whether radar reflectivity is calculated in Meso-NH or inside CoCoMET.

It will be calculated "inside CoCoMET."

L161 Briefly explains the rationale for selecting 30, 40 and 50 dBZ thresholds.

The thresholds provided are examples chosen specifically to reflect the works cited in the text. These thresholds can vary depending on the type of convection, the climate conditions in which the convection occurs, and the purpose of the study. These thresholds are customizable by users and the specific values quoted here simply illustrate an example rather than suggesting threshold values for users.

L170 – Please replace 'recent updates' with a version-specific reference." In 10 years from now, maybe the updates would not be recent anymore.

We removed 'recent updates.'

L189 and L206. Clarify what "types of model outputs" means: file formats, variable sets or both? The three weather models you are describing in the paper?

We rewrote: "*This tracker is currently integrated with only three model outputs (i.e., WRF, RAMS, Meso-HN) in CoCoMET, …*"

Section 2.3 (Output unification). State which new diagnostics are unique to CoCoMET and which already exist in individual trackers.

The following properties are unique to CoCoMET: 2D Feature Perimeter, Feature Surface Area, Feature Irregularity, Maximum height of a Variable of Interest, Cell growth and dissipation rates, Feature Convexity and Sphericity, Linking to ARM Datasets, Calculation and linking to sounding data. Feature area and volume are common outputs from other trackers. We added these to Section 2.5 in the revised manuscript.

Provide a brief explanation of 2-D vs 3-D features for non-experts.

"Features" refer to a single object identified at a single time step. Since the spatial domains of the input datasets can vary, features identified within 3D domains are 3D features and features identified within 2D domains are 2D features. We added this info to Table 2.

L209- Are the deep convective cell/MCSs trackers really sharing outputs? Or are they just similar?

No outputs are shared between trackers within CoCoMET. We are referring to output variables that are common across different trackers. This includes certain cell properties such as feature ID, cell ID, time, and location. However, each tracker may include different or additional outputs or calculate certain properties in distinct ways. To address this variability, we developed an analysis module that allows users to compute additional variables as needed, as described in section 2.3.

We rewrote the sentence: "*Some variables are commonly provided as output by different trackers. These variables include…*"

L221-223. Please explain the new merging/splitting method, or refer to section 2.4.

We rewrote the sentence: "*CoCoMET incorporates a newly developed method for identifying merging and splitting events in both 2D and 3D tracking (see Section 2.4), addressing a key limitation in many existing trackers.*"

L252-253. This is nice. I want to know more examples for other features.

We added some more citations.

Equation 1. V and A are introduced without their subscripts. Consider adding the subscripts in the text.

We removed the subscript 'P' from the equation.

Sec 2.3.6. Clarify whether 3-D features are reconstructed from stacked 2-D features.

This is not the case. Due to different methodologies across trackers for filtering or smoothing the input data during feature identification, the position of a 3D feature will be different from the

position of multiple stacked 2D features. Stacking segmented 2D features wouldn't result in one big 3D feature the same as if it were identified in 3D space.

Section 2.4 (title and content). Define "merge" and "split" early in the subsection title or first sentence. Mergers and splits are not defined (only later L315) and feel like jargon.

*We added these sentences to the beginning of section 2.4: "A merger (or merging event) occurs when two or more individual tracked cells come together and are subsequently identified as a single cell in the next time step. A split (or splitting event) occurs when a single tracked cell divides into two or more distinct cells in a later time step."*

Explain all free parameters (20 % perimeter, 110 % search radius, 50 % overlap, look-ahead of 2 time-steps) and discuss sensitivity.

The parameters can be split into two primary steps: the identification and reconstruction of features, and the classification of events as a merge or split. Given a feature pair (any set of two distinct features we are examining), the perimeter refers to the minimum percentage of the border that must be shared by the features for them to be considered potential merges/splits–this is considered to be the maximum shared border. For instance, if a large cell merges into a small cell-prior to the complete merging–the small feature may share a significant percentage of its edge with the large feature, whereas the large feature may only share a small percentage, thus the maximum of the two values is considered.

We did a sensitivity test below on KVNX NEXRAD radar reflectivity on July 11, 2023 (Table below), differing only one parameter at a time from the default established in Hahn et al. (2025). In many instances, the output is somewhat robust to changes in the perimeter parameter, but increasing it gives rise to less identifications since cells do not often share a significant portion of their borders–this arises from their somewhat elliptical shape.

Once potential merge/split pairs are identified using the perimeter parameter, each cell is individually "reconstructed" independent of any other feature using equation (2) in the manuscript. This is important since the segmentation algorithms do not allow two features to occupy the same space, thus it is necessary to re-segment each feature independently, and adding the search radius parameter allows us to impose physically informed constraints on this segmentation.

Thus, to put bounds on the potential size of each feature, we impose a maximum search radius for which grid cells may or may not be a part of the feature–this initial search radius is defined by making a circular assumption on the feature morphology. However, we recognize that this assumption may not truly capture the spatial extent of a cell, thus we allow the expansion of the radius beyond what is assumed by introducing the search radius adjustment parameter. The product of this parameter and the initial radius defines the search area used to reconstruct the feature.

Now that potential merge/split pairs are identified and cells re-segmented, we use an overlapping technique similar to Feng et al. (2023) and Raut et al. (2021). Between two-time steps, if the two re-segmented features do not share at least overlap% of their area, we drop them from consideration–this can be adjusted to account for sparser temporal information or

faster moving features. In our sensitivity test (Table below), we can see that the overlap% does not play a significant role for slow moving cells as the ones presented here.

Up until this point, the process for merging and splitting identification have been identical as they all deal with spatial extent, however, the look-ahead (or behind) of 2 time-steps parameter is the final condition for merging or splitting, as this is where temporal information is applied. We consider the case of merging as splitting works conversely. So, given a feature pair at a certain time step, we look forward two-time steps. If only one of the two features associated cell IDs' persist, whilst the other does not, we classify this as a merge. In the other cases where both cell IDs persist or neither do, we do not classify this as a merge. This allows the classification to be robust to cell misidentification or "missing" features on a cell track (similar to the use of the "memory" parameter in *tobac*). Our sensitivity test reveals that this parameter has the greatest effect on merge identification since small, short-lived cells arising from misidentification will often be erroneously tracked as merges and other cells may not persist due to shorter lives or coarser temporal resolutions.

| Number of Merges | Perimeter | Search Radius | Overlap | Steps Forward |
|---|---|---|---|---|
| 0 | 30% | 1.1 | 50% | 2 |
| 1 | 20% | 1.1 | 50% | 2 |
| 1 | 10% | 1.1 | 50% | 2 |
| 1 | 5% | 1.1 | 50% | 2 |
| 1 | 20% | 1.2 | 50% | 2 |
| 1 | 20% | 1.3 | 50% | 2 |
| 1 | 20% | 1.5 | 50% | 2 |
| 1 | 20% | 2 | 50% | 2 |
| 1 | 20% | 1.1 | 40% | 2 |
| 1 | 20% | 1.1 | 20% | 2 |
| 1 | 20% | 1.1 | 10% | 2 |
| 1 | 20% | 1.1 | 75% | 2 |
| 0 | 20% | 1.1 | 50% | 2 |

| | | | | |
|---|---|---|---|---|
| 4 | 20% | 1.1 | 50% | 1 |
| 0 | 20% | 1.1 | 50% | 3 |

*We've added this to the Appendix.*

L322- pi should be written with its symbol here. "rsearch", consider subscripting the "search".

*We wrote the symbol pi. The "search" has already been subscripted in the original manuscript and should be represented more accurately in the typeset version of the manuscript*

Equations 2 and 3. The symbol V is reused with a different meaning; please adopt a new symbol for the background threshold, and clarify square root of 2 or of 2 times r_search.

*We changed V to A in the revised version. In addition, we mean the square root of 2 only, multiplied by r_search. This is because the maximum distance a point can have from the center of a square is the square root of 2, multiplied by half the side length of the square.*

Equation 3: same comments as for equation 2.

*We changed V to A in both Equations 2 and 3.*

L358: This forward-looking sentence belongs in the Introduction.

*We moved this sentence to the introduction.*

L369. Specify what the third dimension represents (vertical coordinate?).

*We rewrote: "$k$ is the vertical dimension."*

Section 2.5

L390- clarify whether CoCoMET currently ingests UAV data or whether this is planned.

*We rewrote: "Such capabilities can easily be extrapolated to include distance from observations collected using mobile platforms like research aircrafts or Uncrewed Aerial Vehicles (UAVs), which will be considered in future development."*

Section 3

The configuration system is not fully clear to me: does CoCoMET simply provide a master file that forwards settings to each tracker's native configuration, or does it translate all tracker parameters into a single, unified schema with consistent key names? If the former is true, please clarify the practical benefit of advertising "one configuration file" and explain how this approach adds value relative to running each tracker individually.

*The configuration system is closer to the latter but may probably be a mix of the two situations stated by the reviewer. Consistent key names are proposed in CoCoMET for part of the setting (e.g., bounds of targeted regions for tracking, input directory, tracking variables), but CoCoMET*

also forwards those original settings from each tracker's native configuration (e.g., threshold setting, max velocity). This design makes it easier for users to accommodate the original settings and user guides of each tracker but also takes the benefit of the CoCoMET uniformity. The practical advantage of using a unified configuration file is that users can run a single line of code in the Python interface to execute multiple trackers simultaneously on multiple input data streams. All input data preprocessing, transformation, and output data unification are handled internally by CoCoMET. In contrast, running each tracker individually would require numerous manual steps to achieve the same result. These are cumbersome steps that CoCoMET automates for the users.

We added these texts in Section 3.

L437. Check date format against journal style.

We changed it to the European date format, Day Month Year.

L439. Use consistent SI notation (e.g. m s⁻¹).

We have revised the text and figures to conform to SI notation.

L446- A few times you are referring to the text that a config file is available in Weiner et al (2025). That would be nice to have the config file either in your repository or to have a brief discussion of what the config file is.

Thank you for this suggestion. All configuration files have been uploaded to our GitHub repository (CoCoMET/examples/example_configs/paper_config_files) and Zenodo where they are publicly available for download. The reference and link are provided below and also referenced in "Weiner et al (2025)".

- Weiner, H., Hahn, T., and WANG, D.: Configuration files for running CoCoMET - Examples, Zenodo, https://doi.org/10.5281/zenodo.15048050, 2025.

L450. "increasing trend", make it intemporal.

We added: "*over the past decade.*"

L455- Since Hahn et al. is still in review, consider placing the figure in supplementary material.

Thank you for this suggestion. This paper has been published since this manuscript was submitted and the reviewer comment was posted. Since that manuscript is available now, we have not made any changes to our manuscript. The article is available here: https://agupubs.onlinelibrary.wiley.com/doi/epdf/10.1029/2024JD042586

L460 and Fig. 8 The caption uses "life-cycle bin" whereas the axis label reads "normalized lifetime". In which unit is the lifetime? Hours?

The unit of lifetime is in time units of minutes. However, we normalize the lifetime into a unitless measurement from 0 (first time of cell identification) to 1 (last time of cell identification) to compare cells with considerably different lifetime values on the same figure. The normalized lifetime is then divided into 5 equal bins, with the first bin (0 on the x-axis of Figure 8) including

the instance of first identification for each cell, and 5 including the instance of final identification of each tracked cell. Depending on the number of time steps that each cell was identified for, this may result in an uneven distribution of data across these bins. However, our previous work has found that that has little impact on the overall tracker statistics (see Gupta et al., 2024 and the interactive discussion therein).

We added this info to the figure caption.

Section 4 (Future development). Sections 4.2 and 4.3 both discuss linking to additional external data sets; consider merging them or clarifying the distinction.

Section 4.2 covers the linkage to the data used as inputs, while Section 4.3 links the tracking results to other datasets during post-processing or within the analysis module.

We have renamed Section 4.2 as "*Enhancing Support for Pre-tracking Input Datasets*" and renamed Section 4.3 as "*Enhancing Support for Post-tracking Analysis Datasets*."

Explain what you mean by "thermodynamic data sets" and by "external" data.

We removed the words 'thermodynamic' and 'external,' as all data are technically external to CoCoMET. These datasets are only external to the trackers or not the input data that were used for tracking.

L492. Check spelling for "PyFLEXTRKR" and cite TempestExtremes.

We changed it to "PyFLEXTRKR," and we have now added the citation for TempestExtremes in Section 4.

The final sentence of Section 4 repeats material already discussed. Please also check "Stage VI" throughout the manuscript.

We changed it to "*stage IV*."

Scaling limitations. Briefly discuss potential bottlenecks for global domains or multi-year archives (memory, I/O, parallel efficiency)

Thank you for the suggestion, as this is a major concern for users. The primary limitations on CoCoMET are built into its dependencies and user equipment. For instance, large netCDF datasets inherently take longer to load into user memory, and datasets which–when aggregated–exceed the total memory capacity of the user's machine will lead to slowdowns or the program not being able to run. However, CoCoMET does take advantage of the multithreading capabilities of packages such as *xarray* when available, resulting in speedups akin to what the respective packages advertise. However, CoCoMET has its own limitations when in the analysis stage of its operation.

A major bottleneck users may see results from the type and setup of their tracking goals. For example, if a user is tracking equivalent radar reflectivity with high thresholds, they may only identify a few cells total and the analysis portion will run quickly. But, if the user is tracking updraft cores with relatively low thresholds on the same dataset, there may be hundreds of

identified objects, resulting in greater computational burden for both the trackers and analysis. Noting that these issues grow with larger domain sizes in the absolute sense–i.e. total number of grid cells, not the geographical or temporal size of the domain.

We added this context to the revised manuscript: "*The performance of CoCoMET is limited by its dependencies, the user's machine, and the user's tracking goals. Initial input and tracking speeds are largely dictated by individual dependency performance, but speedups–such as multithreading from dask–are used when offered. Large datasets may be exceptionally slow, or even fail to run, if the user's machine does not have sufficient memory or other processing power. However, it is often the case that the tracking setup itself is the issue. For instance, the tracking of cell updrafts–where hundreds or thousands of features may be identified–are going to slow down the analysis module of CoCoMET significantly due to the high computational complexity of the analysis algorithms. We recommend running CoCoMET, initially, without the analysis module to ensure your configuration parameters are set correctly and there are not large amounts of undesired features.*"

Fig.2 What does the grey arrow mean? The top ones look like they are mentioning preprocessing or liking data, and the bottom ones look like they are for running the tracker. Is the big blue arrow for mentioning the output of CoCoMET? It would be nice to see what are inputs and what are outputs.

The grey arrows on the original plots do not have particular meanings. We made them all grey in the revised figure below to avoid ambiguity. We also added a sector about the outputs (pink boxes). Inputs are illustrated in the blue boxes.

[Figure]

Fig. 3. It would be nice to see a feature of convexity = 1 and a feature is convexity very close to 0.

A feature with convexity = 1 would represent a perfect circle, which does not exist in the real-world data, as the objects are usually presented with grids. A feature with convexity = 0 does not really exist either. The examples presented in Figure 3 are the max and min convexities we can find in our example datasets.

Fig. 4. explain the colour scheme.

The explanation for the colour scheme is added.

Figures 5–6 (merge/split examples) lack variable names, thresholds

Variable names and thresholds are added.

Fig. 7. Caption: "MesoHN" → "Meso-NH" and unify axis units.

We made the changes accordingly.

Fig. 8. Observed and simulated, please specify which one is which.

We modified the caption: "*Box-whisker plot of tracked cell properties (both observed [NEXRAD] and simulated [RAMS]) as a function…*"

Fig. 9. maybe add a tick for each hour for clarity.

We added ticks.

References: Check DOI formatting; several have duplicated "https://doi.org/https://doi.org".

We checked the duplicated doi urls and fixed them in the revised manuscript.

Replace straight quotes with typographic quotes, and use consistent en-/em-dashes.

Modified.

**Reviewer #2**

The manuscript at hand introduces a new python-based open-source software tool that allows for the application of multiple algorithms that identify and track convective clouds and storms in diverse datasets. The software tool primarily standardizes the input and output data to facilitate a workflow where multiple tracking algorithms are applied to datasets in different data formats. In addition, the tool offers new features and enhances thereby the capabilities of existing tracking tools. These new features include, for instance, the computation of additional storm characteristics such as cell growth and dissipation rates, convexity and irregularity as well as the implementation of a novel method for merging and splitting.

First of all, I would like to apologize for my delayed evaluation. This project uses state-of-the-art software tools and standards, follows open-source and open science principles and addresses a major challenge in weather and climate science data: How to best unify the different data formats and tools that we have available to enable more systematic data analyses of observations and models. The paper is interesting and generally well-organized. The software package is explained in a clear manner, including examples for analyses and applications. I will recommend this paper for publication after my comments have been addressed. I see the need for clarifications at some locations in the text. In particular, the potential for enhancements and how to add new datasets need to be discussed more clearly.

General comments

Introduction - I think the introduction could be improved by mentioning a few studies that have done model and dataset intercomparisons in a more complicated way. This is to highlight and better explain what kind of studies would benefit from the presented framework.

Thank you for this suggestion. We have added the following text to the introduction:

*"The use of object tracking for model evaluation is gaining popularity, both for model intercomparisons and for comparisons between models and observational data (e.g., Prein et al., 2024; Feng et al., 2025; Gilmour et al., 2025; Hahn et al., 2025). Feng et al. (2025) evaluated various the DYnamics of the Atmospheric general circulation Modeled On Non-hydrostatic Domains (DYAMOND) model simulations of tropical MCS against satellite precipitation and brightness temperature products by using multiple different trackers. They reported that while the frequency of observed MCSs can have a spread of a factor of 2-3 across trackers, robust model evaluation can be achieved despite differences in the formulation of different trackers. In another tracker intercomparison, Prein et al. (2024) examined the sensitivity of MCS statistics from climate model simulations to the formulations of six different trackers. This work showed the use of different trackers can influence the conclusions drawn while evaluating model simulations against observations and that the frequency, size, and duration of tracked MCSs are highly susceptible to the tracker being used despite the use of consistent criteria to define an MCS."*

Generalizability - I can truly see the challenge that is addressed with this project, since running multiple trackers on multiple datasets can be cumbersome. However, it remains unclear to me how abstract and modular this framework is actually implemented such that new trackers and datasets can be easily added. Could you clarify what the pre-processing steps would be to add new datasets and be a bit more general of what the data structure needs to be. Appendix A addresses this partly, but I am not sure whether there is some flexibility in the variable names, etc. My understanding is also that there would be a tradeoff between a generalizable approach that focuses on using a specific tracker like MOAAP on any dataset (the focus would be on making all data formats and variables work with MOAAP in a pre-processing step) vs. making all trackers compatible with a certain dataset. What is the best approach for an abstract implementation of this?

CoCoMET has the flexibility to track any native variable that the trackers were designed to track. We do not put any restrictions there. Generally speaking, the dataset should be structured on a regular grid or be convertible to a gridded format (CoCoMET handles this conversion for NEXRAD). It must also include a time dimension. From a scientific perspective, certain variables may be more suitable for tracking with specific trackers—for example, using reanalysis data with MOAAP to track atmospheric rivers. Ultimately, it is up to the user to decide whether the tracked features have scientific relevance. We only provide the option for evaluation or comparison.

For a newly implemented dataset, a series of files facilitating the usage of each currently implemented tracker on that dataset can be created for the unique conversions required to run those trackers on the dataset. This method eliminates the need for any prerequisites of new datasets. Technically speaking, CoCoMET handles all of the tracker/dataset pairs uniquely, and there is no preference toward a single tracker or dataset, and the user is free to perform a large variety of analyses with ease and to decide which tracker to use or to not use.

To add an additional dataset into CoCoMET, contributors should follow these steps:

1. Update User Interface Layer: Create a run_<data_name> function in user_interface_layer.py which handles the calling of all possible trackers for the new dataset. Then follow the pre-existing procedure for other datasets in the functions CoCoMET_start, CoCoMET_start_multi, and CoCoMET_load.

2. Calculate Data Variables: Create a <data_name>_calculate_products.py file to facilitate the calculation of DBZ, WA, TB, and PR if the variables do not already exist in the data.

3. Generate Iris Cube: Create <data_name>cube.py to facilitate generation of an iris Cube for one data variable.

4. Create Load File: Create a <data_name>_load.py file to load the data into an xarray Dataset and reference the iris Cube generator and data variable calculation files.

5. Run Individual Trackers: Create <data_name>_<tracker>.py files for each possible tracker which can be used on the data to facilitate tracker parameterizations.

6. Reference New Files in Wrapper: Finally, import newly created files and all necessary functions in  run_tracker_wrapper.py and ensure proper naming conventions of each file.

We added these steps to section 4.2.

Global vs. regional - Related to the question above, are there any restrictions on regional vs. global datasets? The analysis and examples seem to be focused on CONUS, and I am curious if one could easily add global models and observational datasets.

It is not restricted to regional data or global data. Any global raw data (simulation and observations) or regional data of another location can work if the datastream is implemented in CoCoMET already. If not implemented already, users can also input a gridded dataset that follows the variable names and structure that CoCoMET allows.

Unstructured grids - How will this package address the challenge of unstructured grids? It is stated in l. 93, that there is a plan to include models such as ICON. Will there be a pre-processing step in which the data are regridded onto a common grid or how will the different grids be handled? I am also curious if the handling of such models will be model-specific or generally applicable such that other models can be easily added (e.g. Model Prediction Across Scales/MPAS which is the successor and replacement of many WRF applications).

Interesting point. In fact, NEXRAD radar data is an unstructured grid, and we use PyART to grid it, so for any additional unstructured grid, we will be doing the same thing: finding a new package or finding a way to grid it before passing it to the trackers.

The regridding will be done in the pre-processing step as either a separate or model specific module in which the data are regridded onto a regular grid for easy comparison with other regular-grid datasets. CoCoMET is flexible to handle both options. A universal function would be prioritized but the final decision may depend also on the computational efficiency, community demands, and the differences in the unstructured data.

Version handling - A general challenge that I see with this package is that it is heavily dependent on the versions of the supported tracking algorithms. For example, tobac just released a new version (v1.6.0) in which xarray is supported eliminating the need for iris. How do you plan to maintain this tool ensuring compatibility with the versions of the tracking algorithms? Are you requiring certain versions of the latter or could, for instance, the latest version of tobac be run with the latest installation of CoCoMET? Do you plan to have any active communication and collaboration with the developers of the supported tracking libraries?

We thank the reviewer for the suggestion. Indeed, this is an important part of this package. We plan to check the release of each incorporated tracker on a regular basis and install the new version of the released tracker in CoCoMET. We have joined the tobac user community and are aware of this new release, although some other packages do not have a user community or formal release notice, in this case, we will contact the developer individually every 6 months to make sure we are up to date.

We added these sentences to the revised manuscript in Section 4.1:"*CoCoMET developers will update the package every 6 months to account for new releases of the existing and newly incorporated trackers. Depending on the number of CoCoMET users, a discussion forum will utilize the open-source ecosystem to incorporate the community's suggestions into any major releases.*"

Unit testing - related to the former comment, it seems like unit testing and continuous integration would be very valuable for this type of package. Otherwise, it could be pretty hard to identify where the code breaks when datasets and the tracking algorithms change.

We thank the reviewer for their suggestion. We have added continuous integration into our github routine checking for formatting with *black*, automatically updating the documentation page with *pdoc*, running the linter, *pylint*, and running our tests automatically with *pytest.* We include a base functional test where we ensure the package runs as desired with a dataset which includes a variety of model outputs and observational data in addition to a "ground truth" output we compare against–allowing us to know CoCoMET maintains core functionality throughout any changes and updates. The *verbose* flag in CoCoMET is set to true for these tests to allow us to identify major breaking points.

Additionally, since many of CoCoMET's features are software layers interfacing with other packages, it would be redundant to implement unit tests to cover all of the inputs and tracking modules, since those would be captured by the packages' respective tests. However, we do find potentially issues in the analysis module, where the potential for error is higher. Hence, we implement unit tests which cover our analysis packages and ensure consistent computation.

We add the following texts to the revised manuscript: "*To ensure the stability and reliability of CoCoMET, we have incorporated continuous integration into our GitHub workflow. This framework includes automated checks for code formatting using Black, documentation updates via pdoc, linting with pylint, and test execution through pytest.*

*As part of our testing strategy, we include a functional test that ensures the package runs correctly using our testing dataset (Hahn, 2025), and a corresponding pre-run "ground truth" output. By comparing CoCoMET's output against this ground truth, we verify that core functionalities are preserved across code updates and changes. To facilitate debugging, the verbose flag is enabled during these tests, allowing clear identification of potential breaking points. In addition, we implement a set of unit tests for the analysis module to ensure accuracy and reproducibility of computed diagnostics.*"

Atmospheric features beyond deep convective systems - From the README file, it looks like TAMS and MOAAP are only supported for MCS tracking. Is that right or is it, for instance, possible to use MOAAP to track atmospheric rivers in WRF model output using the current version of CoCoMET?

Although most of the contexts and examples are about convective clouds in the manuscript, CoCoMET does output the original outputs of each tracker. These outputs are just not part of the CoCoMET standardized output. For example, if a user wants to track atmospheric

rivers through MOAAP, they just need to specify a savepath for MOAAP specific outputs in the CONFIG file, and those outputs will be saved.

Documentation - Since the python package has no formal documentation page, I think it is important to mention the user guide. I strongly recommend filling in the section of "how to set up a config file" because that is still not very easy to figure out from the examples.

Thank you for the suggestion. We have updated the landing page of CoCoMET's GitHub (https://github.com/ASCENT-BNL/CoCoMET/tree/test_branch) to include installation instructions, a link to the user guide PDF, links to the example Jupyter notebooks, and an example CONFIG.yml file. Additionally, we have set up a formal documentation page using *pdoc* at https://ascent-bnl.github.io/CoCoMET. The link to the website can also now be found on the GitHub landing page. How to set up a configure file is detailed in this document: https://github.com/ASCENT-BNL/CoCoMET/blob/master/docs/user_guide/cocomet_user_guide.pdf

Merging and splitting - Is it possible to use the suggested framework to compare the merging and splitting capabilities of MOAAP, TAMS, tobac as well as the novel method?

Tobac includes its own method for detecting merging and splitting events and can output this information. In contrast, MOAAP and TAMS state that such events can be detected but do not currently provide this information in their outputs. Therefore, in the comparison below, we evaluate our algorithm only against the merging and splitting detection in Tobac.

Tobac applies user-defined thresholds for maximum allowed spatial and temporal separations between cells to be considered potential mergers/splits. This is then used in an implementation of Kruskal's algorithm to construct a minimum Euclidean distance spanning tree. This tree structure, formed over spatial and temporal domains, is what defines the merge or split events. In contrast, CoCoMET uses multiple criteria to identify cell mergers/splits, such as the percentage of spatial overlap between features and whether features share edges (see responses to the last question of reviewer 1 on page 9 for more info on each parameter).

Unfortunately, there are no directly comparable thresholds or parameters between the two algorithms that would allow for a fair comparison. Therefore, we performed our comparison using several settings for CoCoMET and tobac (Figures below). It's important to note that these thresholds should be tailored to the specific application, as appropriate settings may differ between isolated and organized convection.

[Figure]

**Figure 1:** The locations and timing (colors) of mergers detected using both tobac (a) and CoCoMET (b) merging algorithms. The input data are infrared brightness temperatures calculated based on WRF simulations at 4 km resolution on June 19, 2013. Cells are tracked using tobac. For CoCoMET: touching_threshold = 0.15, score_weight_1 = 1.5. For tobac: merge_dist = 25 (100 km), frame_len = 7.

[Figure]

**Figure 2**: Same as Figure 1, but For CoCoMET: touching_threshold = 0.15, score_weight_1 = 1, overlap_threshold = 0.2. For tobac: merge_dist = 25, frame_len = 3.

Detailed comments

78 - maybe worth mentioning that this follows tobac's nomenclature (see e.g., Sokolowsky et al., 2024)

We note this in our updated text as follows:

*"This is consistent with the terminology used within tobac (e.g., Sokolowsky et al., 2024)."*

164: The segmentation module of tobac enables the spatial definition of the identified or tracked objects and the calculation of bulk statistics of each object. However, it should be highlighted that the segmentation procedure can be done based on the output of the feature detection (as a second step, as currently noted in the manuscript) but also based on the output of the tracking procedure (based on the cell locations as a third step). In addition, users could also decide to not apply the segmentation module, as this is not a required step. This may be useful and save a lot of computation time and data storage when the user mainly needs the time and locations of tracked cells.

We have added the following text:

*"tobac is flexible as feature segmentation can be completed after either the identification or linking step, and can be bypassed if a user does not need spatial information for the tracked features."*

156: It could be useful to highlight here that both of the implemented trackers as well as the trackers that you plan to implement are all members of the MCSMIP intercomparison (Feng, Z., Prein, A. F., Kukulies, J., Fiolleau, T., Jones, W. K., Maybee, B., ... & Mejia, J. F. (2025). Mesoscale convective systems tracking method intercomparison (MCSMIP): Application to DYAMOND global km-scale simulations. Journal of Geophysical Research: Atmospheres, 130(8), e2024JD042204.).

We moved this line to Section 4 in response to another reviewer comment. At that location, we have added the following text:

*"These trackers, along with the trackers already supported by CoCoMET, were included in a recent MCS Tracking Method Intercomparison (Feng et al., 2025)."*

427: Does this standard output follow tobac output structure or is it different? And do you leave it up to the user to save this in whatever file format they want or is there a standard for this, too?

Our output formatting procedure is similar to tobac but with distinct differences. For example, we provide the output as either a geopandas dataframe or an xarray dataset, which can easily be saved as either a .csv file or a netCDF file. Eventually, the user has an option to save the output file in any format they prefer. We have added a routine to save it as a netCDF file at 'CoCoMET/examples/saving_data_example.ipynb'.

Fig. 9 - To understand the differences between the trackers in this specific example, could you please clarify what area and lifetime minimum you have chosen for the different trackers and if the three trackers offer the same initial criteria. I am wondering, for example, if the reason for MOAAP showing significantly less cells is that it has by default a more strict requirement of grid cell connectivity and spatial continuity, whereas you can set a minimum area for each object and thresholds in tobac.

Thanks for the suggestion. We did not apply thresholds for minimum area or lifetime in the original Figure 9. This was intentional, as the figure was meant to illustrate the inherent characteristics of each tracker based on their original design. We believe this information is important for guiding users in selecting the appropriate tracker for their research purposes. For example, based on the figure, it is evident that *tobac* is well-suited for tracking isolated convection, while the other two trackers are designed for identifying MCSs.

We also performed a sensitivity test by limiting tracked cells to a minimum area of 4,000 km² and a minimum lifetime of 2 hours (figure below). We found that the number of cells tracked by *tobac* decreased significantly. Note that, as other trackers have additional criteria for cell identification, these results are not entirely comparable even when lifetime and minimum area are fixed.

We added these texts to the revised manuscript.